# AUTONOMOUS CATHETERIZATION WITH OPEN-SOURCE SIMULATOR AND EXPERT TRAJECTORY

## ABSTRACT

Endovascular robots have been actively developed in both academia and industry. However, progress toward autonomous catheterization is often hampered by the widespread use of closed-source simulators and physical phantoms. Additionally, the acquisition of large-scale datasets for training machine learning algorithms with endovascular robots is usually infeasible due to expensive medical procedures. In this paper, we introduce CathSim, the first open-source simulator for endovascular intervention to address these limitations. CathSim emphasizes real-time performance to enable rapid development and testing of learning algorithms. We validate CathSim against the real robot and show that our simulator can successfully mimic the behavior of the real robot. Based on CathSim, we develop a multimodal expert navigation network and demonstrate its effectiveness in downstream endovascular navigation tasks. The intensive experimental results suggest that CathSim has the potential to significantly accelerate research in the autonomous catheterization field. Our project is publicly available at https://anonymous.4open.science/r/cathsim-E168.

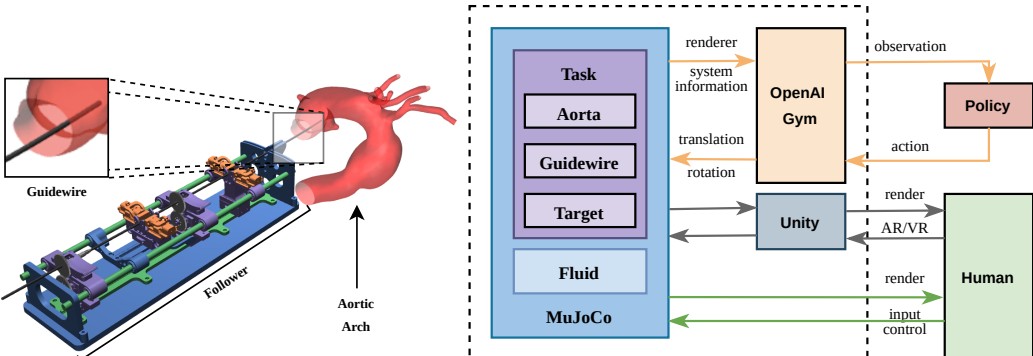

Figure 1: An overview of CathSim.        Figure 2: The design architecture of CathSim.

## 1 INTRODUCTION

Endovascular interventions are commonly performed for the diagnosis and treatment of vascular diseases. This intervention involves the utilization of flexible tools, namely guidewires, and catheters. These instruments are introduced into the body via small incisions and manually navigated to specific body regions through the vascular system (Wamala et al., 2017). Endovascular tool navigation takes approximately 70% of the intervention time and is utilized for a plethora of vascular-related conditions such as peripheral artery disease, aneurysms, and stenosis (Padsalgikar, 2017). Furthermore, they offer numerous advantages over traditional open surgery, including less recovery time, minimized pain and scarring, and a lower complication risk (Wamala et al., 2017). However, surgeons rely on X-ray imaging for visual feedback when performing endovascular tasks. Thus, they are overly exposed to operational hazards such as radiation and orthopedic injuries. In addition, the manual manipulation of catheters requires high surgical skills, while the existing manual solutions lack of haptic feedback and limited visualization capabilities (Omisore et al., 2018).

Recently, several robots have been developed to assist surgeons in endovascular intervention (Kundrat et al., 2021). This allows surgeons to perform endovascular procedures remotely (Mahmud et al., 2016). However, most of the existing robotic systems are based on the follow-the-leader (master-slave) convention, wherein navigation is still *fully reliant* on surgeons (Püschel et al., 2022). Furthermore, the use of manually controlled robotic systems still requires intensive focus from the surgeon, as well as a prolonged duration compared to its non-robotic counterpart (Jara et al., 2020). We believe that to overcome these limitations, it is crucial to develop autonomous solutions for the tasks involved in endovascular interventions.

In this paper, we introduce CathSim, a significant stride towards *autonomous catheterization*. Our development of this open-source endovascular simulator targets the facilitation of *real-time training of machine learning algorithms*, a crucial advancement over existing, often closed-source simulators burdened with computational demands (See et al., 2016). CathSim distinguishes itself by focusing on machine learning applicability, thereby overcoming common design restrictions found in other simulators (Table 1). It boasts features essential for rapid ML algorithm development, such as easy installation and gymnasium support, anatomically accurate phantoms including high-fidelity aortic models from Elastrat Sarl Ltd., Switzerland, and a variety of aortic arch models for extensive anatomical simulation. CathSim also achieves high training speeds, balancing computational demand and efficiency, and integrates advanced aorta modeling with detailed 3D mesh representations for realistic simulations. Additionally, it offers realistic guidewire simulation and compatibility with AR/VR training through Unity integration, enabling advanced surgical training applications. Moreover, CathSim facilitates targeted algorithm development for specific aortic complications, thereby enhancing the effectiveness of medical interventions. These features collectively position CathSim as a versatile and invaluable asset in both surgical training and the development of groundbreaking machine learning algorithms within the medical community.

Recognizing that autonomous catheterization is an emerging task within the machine learning domain, we introduce an expert trajectory solution as a foundational baseline. These expert trajectories model complex surgical procedures, offering a rich, practical learning context for developing autonomous systems (Kiran et al., 2021). By enabling observational learning, these systems can adeptly mirror expert maneuvers, significantly reducing the learning curve from novice to skilled interventionists. CathSim's risk-free, diverse, and dynamic simulation environment allows autonomous systems to iterate and refine their performance safely, informed by expert actions (Li et al., 2022). Our research demonstrates that leveraging expert-guided learning in a simulated setting markedly enhances the effectiveness of downstream ML tasks in autonomous catheterization, such as imitation learning and force prediction. The contributions of this work are twofold:

- Introduction of CathSim, an innovative, open-source endovascular navigation simulator, specifically engineered for autonomous catheterization. It features real-world emulation, realistic force feedback, rapid training capability, and is AR/VR ready, making it an essential asset for the ML community in medical simulations.
- Development of an expert trajectory network, along with a novel set of evaluation metrics, to demonstrate its efficacy in pivotal downstream tasks like imitation learning and force prediction, thus pushing the boundaries of ML in autonomous medical interventions.

## 2 RELATED WORK

**Endovascular Simulator.** Research on simulators for minimally invasive surgery categorizes the simulation level into four distinct categories: synthetic, virtual reality, animal, and human cadaver (Nesbitt et al., 2016). Each type of simulation environment possesses unique advantages and limitations, as detailed in numerous studies (Dequidt et al., 2009; Talbot et al., 2014; Sinceri et al., 2015). The primary focus of these environments lies in trainee skills' development (Nesbitt et al., 2016; Talbot et al., 2014), path planning (Dequidt et al., 2009), and the enhancement of assistive features, such as haptic feedback (Molinero et al., 2019). Recently, the use of synthetic simulators, such as high-fidelity phantoms, has been investigated through the application of imitation learning techniques (Chi et al., 2020). Simultaneously, other studies have utilized simulation environments and tested their models on bi-dimensional synthetic phantoms (Faure et al., 2012; Lillicrap et al., 2015; Dequidt et al., 2009; Talbot et al., 2014; Wei et al., 2012). Nevertheless, despite advancements, challenges persist due to the physicality, real-time-factor, or closed-source nature of the simulators.

Table 1: Endovascular simulation environments comparison.

| Simulator | Physics Engine | Catheter | AR/VR | Force Sensing | Open-source |
|---|---|---|---|---|---|
| Molinero et al. (2019) | Unity Physics | Discretized | ✗ | Vision-Based | ✗ |
| Karstensen et al. (2020) | SOFA | TB theory | ✗ | ✗ | ✗ |
| Behr et al. (2019) | SOFA | TB theory | ✗ | ✗ | ✗ |
| Omisore et al. (2021) | CopelliaSim | Unknown | ✗ | ✗ | ✗ |
| Schegg et al. (2022) | SOFA | TB theory | ✗ | ✗ | ✗ |
| You et al. (2019) | Unity Physics | Discretized | ✗ | Vision-Based | ✗ |
| **CathSim** (ours) | MuJoCo | Discretized | ✓ | ✓ | ✓ |

Table 1 shows a comparison of current endovascular simulators. Unlike other simulators, CathSim provides an open-source environment that is well-suited for training autonomous agents. Built on MuJoCo (Todorov et al., 2012), CathSim offers real-time force sensing and high-fidelity, realistic visualization of the aorta, catheter, and endovascular robots. In practice, CathSim can be utilized to train reinforcement learning (RL) agents or serve as a skill training platform for interventionists.

**Autonomous Catheterization.** The advancement of machine learning has paved the way for initial results in autonomous catheterization. While initial research primarily concentrates on devising supportive features (You et al., 2019), an evident shift towards higher degrees of autonomy has emerged, such as semi-autonomous navigation (Yang et al., 2017). Several studies within this domain have employed deep RL techniques (Behr et al., 2019; Karstensen et al., 2020; Kweon et al., 2021; Athiniotis et al., 2019; Omisore et al., 2020; 2021), typically exploiting images obtained during fluoroscopy (Ji et al., 2011). Nonetheless, a number of approaches do not depend on RL. For instance, several works (Qian et al., 2019; Cho et al., 2021; Schegg et al., 2022) have utilized the Dijkstra algorithm (Dijkstra, 1959), following a centerline based navigation paradigm. A different approach involves the use of breadth-first search (Fischer et al., 2022). Despite these promising results, a significant portion of the research is still positioned at the lower end of the autonomy spectrum (Yang et al., 2017), primarily relying on physical or closed-source environments.

**Imitation Learning.** Recent advancements in RL have enabled imitation learning to be accomplished based on human demonstration (Ho & Ermon, 2016). This is especially beneficial for tasks requiring complex skills within dynamic environments, such as surgical tasks within evolving anatomies. Imitation learning frameworks have already been successfully deployed in executing real-world tasks via robotic systems, such as navigation and manipulation (Tai et al., 2018; Finn et al., 2016). Learning-based methods on demonstration have been employed in several studies within the field of endovascular navigation (Rafii-Tari et al., 2014; 2013; Chi et al., 2018b;a; 2020). These have been paired with the incorporation of hidden Markov models or dynamical movement primitives (Saveriano et al., 2021), while recent works use generative adversarial imitation learning (Ho & Ermon, 2016). By utilizing insights from deep RL, the level of surgical autonomy could potentially evolve towards task autonomy, wherein the robot, under human supervision, assumes a portion of the decision-making responsibility while executing a surgical task (Dupont et al., 2021).

## 3 THE CATHSIM SIMULATOR

Fig. 1 and Fig. 2 shows the overview and system design of CathSim with four components: *i)* the follower robot, as proposed by Abdelaziz et al. (2019), *ii)* the aorta phantom, *iii)* the guidewire model, and *iv)* the blood simulation and AR/VR. We choose MuJoCo as our foundation platform for two reasons: First, MuJoCo is computationally efficient, making it an ideal choice for fast development. Second, MuJoCo is well integrated with the machine learning ecosystem, offering researchers a familiar interface and accelerating algorithm development to address endovascular intervention challenges. Since there are several methods to simulate each component in our system, it is a challenging task to find the optimal combination. We design our system such that it is modular, upgradable, real-time, and extendable. Please see the Appendix A for more design information.

**Simulation Model.** Although our CathSim has several components, we assume that all components are built from rigid bodies (instead of soft bodies). This is a well-known assumption in many state-of-the-art simulators to balance the computational time and fidelity of the simulation (Faure

et al., 2012; Todorov et al., 2012). We employ rigid bodies, governed by the general equations of motion in continuous time, as follows:

$$M\dot{v} + c = \tau + J^T f \ . \tag{1}$$

where $M$ denotes the inertia in joint space, $\dot{v}$ signifies acceleration, and $c$ represents the bias force. The applied force, $\tau$, includes passive forces, fluid dynamics, actuation forces, and external forces. $J$ denotes the constraint Jacobian, which establishes the relationship between quantities in joint and constraint coordinates. The Recursive-Newton-Euler algorithm (Featherstone, 2014) is employed to compute the bias force $c$, while the Composite Rigid-Body algorithm (Featherstone, 2014) is used to calculate the joint-space inertia matrix $M$. Forward kinematics are utilized to derive these quantities. Subsequently, inverse dynamics are applied to determine $\tau$ using Newton's method (Todorov, 2011).

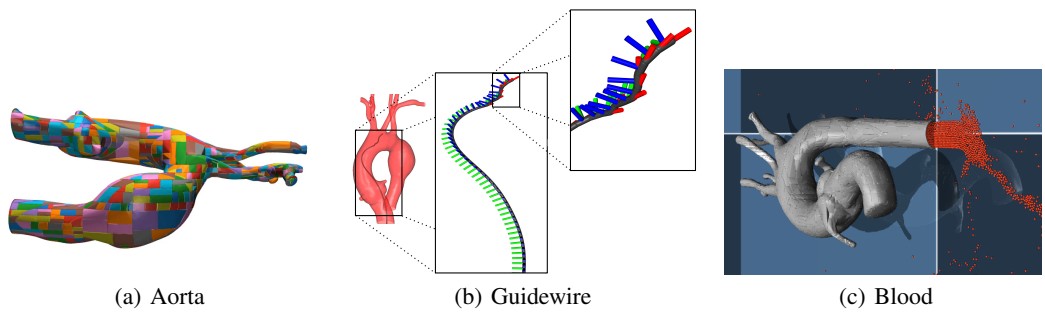

(a) Aorta          (b) Guidewire          (c) Blood

Figure 3: The visualization of the aorta, guidewire and blood in our simulator.

**Aorta.** We scan four detailed 3D mesh representations of aortic arch models, which are created using clear, silicone-based anthropomorphic phantoms (manufactured by Elastrat Sarl Ltd., Switzerland). This is followed by the concave surface decomposition into a collection of near-convex shapes using the volumetric hierarchical approximate decomposition (V-HACD) (Mamou & Ghorbel, 2009), resulting in a set of convex hulls. These convex forms are subsequently incorporated into our simulator for collision modeling. Their use significantly simplifies computations (Jiménez et al., 2001) and allows for the implementation of multipoint contacts using the MuJoCo (Todorov et al., 2012). The combination of these steps results in our simulated aorta, as depicted in Fig. 3(a).

**Guidewire.** A rope-like structure designed to direct the catheter towards its intended position. The guidewire is composed of the main body and a tip, where the tip is characterized by a lower stiffness and a specific shape (depending on the procedure). Modelling the flexibility of a guidewire is accomplished by dividing it into many rigid components linked together by revolute or spherical joints (Burgner-Kahrs et al., 2015). This form of representation has been proven to confer accurate shape predictions (Shi et al., 2016), while characterized by a low computational cost compared to its counterparts (Burgner-Kahrs et al., 2015). To ensure real-time functionality during simulations, we developed a serpentine-based model comprising numerous rigid segments interconnected by revolute joints that approximate the continuous contour and bending behavior of the guidewire. The collision properties of the guidewire's segments comprise a capsule-based shape, composed of a cylindrical core flanked by conical terminations designed as opposed to hemispherical caps. The caps merge along their common interface, forming the wire's exterior surface. This design mimics the motion and shape of the real catheter (Fig. 3(b)).

**Blood Simulation.** Although blood modeling is not the primary focus of our current work, for reference purposes, we include a basic implementation. Our model treats blood as an incompressible Newtonian fluid, following the real-time methodology described in the study by Wei et al. (2012) (see Fig. 3(c)). We intentionally omit the dynamics of a pulsating vessel, resulting in the simplification of assuming rigid vessel walls. This simplification is a common approach in the field, as seen in works like Yi et al. (2018), Behr et al. (2019), and Karstensen et al. (2020), and it helps to minimize computational demands while effectively simulating the forces opposing guidewire manipulation.

# 4 AUTONOMOUS CATHETERIZATION WITH EXPERT NAVIGATION NETWORK

Inspired by autonomous driving and human-robot interaction field (Kiran et al., 2021; Kim et al., 2021), we develop an expert navigation network for use in downstream tasks. We use CathSim to generate a *vast amount of labeled training samples*, enabling our model to learn from diverse scenarios. By exposing the model to different scenarios, we can enhance its ability to generalize to real-world situations (Zhao et al., 2020). Furthermore, we also *leverage additional information* that is unavailable within the real systems (Püschel et al., 2022), such as force (Okamura, 2009) or shape representation (Shi et al., 2016), to further enhance our expert navigation system. We note that our simulator offers a wide range of modalities and sensing capabilities compared to the real-world endovascular procedure where the sensing is very *limited* (the surgeons can only rely on the X-ray images during the real procedure). By combining these modalities, we aim to develop an expert navigation system that not only achieves high performance but also ensures sample efficiency.

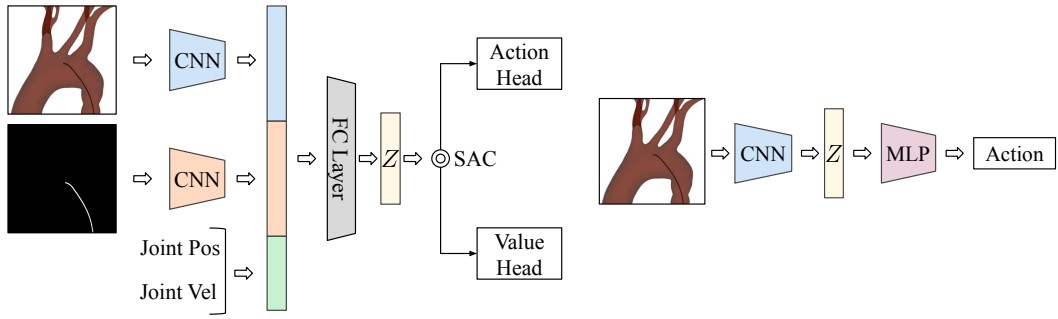

Figure 4: The expert navigation network.  Figure 5: Downstream imitation learning.

## 4.1 EXPERT NAVIGATION NETWORK

Our Expert Navigation Network (ENN) is a *multimodal* network trained on CathSim. Firstly, we include a semantic segmentation of the guidewire as one of the modalities. This allows the expert to accurately perceive the position and shape of the guidewire during navigation, enabling safe movements within the blood vessels. Secondly, we set joint position and joint velocity values for the guidewire. By incorporating these data, we can formulate the guidewire's kinematics and dynamics details (Tassa et al., 2018), thus allowing for more coordinated and efficient navigation compared to previous works (Rafii-Tari et al., 2012; Song et al., 2022). Thirdly, we include the top camera image as another input modality. This visual input provides contextual information about the surrounding environment, enabling the expert to make informed decisions based on the spatial arrangement of blood vessels and potential obstacles (Cho et al., 2021).

We employ Convolutional Neural Networks (CNN) to extract visual features from the input images and the segmentation map, and a Multi-Layer Perceptron (MLP) to process the joint positions and joint velocities. The resulting feature maps are then flattened and concatenated. A fully connected layer is then used to map the features to the feature vector $Z$. By combining these modalities, our expert navigation system can learn the complex mapping between inputs and desired trajectories. The feature vector $Z$ serves as the input for training the soft-actor-critic (SAC) policy $\pi$ (Haarnoja et al., 2018), a core component of our reinforcement learning approach. The overall ENN architecture is visualized in Fig. 4 and the detailed implementation can be found in Appendix E.

## 4.2 DOWNSTREAM TASKS

We demonstrate the effectiveness of the ENN and our CathSim simulator in downstream tasks, including imitation learning and force prediction. Both tasks play an important role in practice, as they provide critical information for the surgeon in the real procedure.

**Imitation Learning.** We utilize our ENN using behavioral cloning, a form of imitation learning (Hussein et al., 2017), to train our navigation algorithm in a supervised manner. This approach emphasizes the utility of the simulation environment in extracting meaningful representations for imitation learning purposes. Firstly, we generate expert trajectories by executing the expert policy, denoted as $\pi_{\exp}$, within CathSim. These trajectories serve as our labeled training data, providing the desired actions for each state encountered during navigation. Secondly, to mimic the real-world observation space, we select the image as the only input modality. Thirdly, we train the behavioral cloning algorithm by learning to replicate the expert's actions given the input observations and optimizing the policy parameters to minimize the discrepancy between the expert actions and the predicted actions:

$$\mathcal{L}(\theta) = -\mathbb{E}_{\pi_\theta}[\log \pi_\theta(a|s)] - \beta H(\pi_\theta(a|s)) + \lambda||\theta||_2^2 \ . \tag{2}$$

where $-\mathbb{E}_{\pi_\theta}[\log \pi_\theta(a|s)]$ represents the negative log-likelihood, averaged over all actions and states; $-\beta H(\pi_\theta(a|s))$ is the entropy loss, weighted by $\beta$ and $\lambda||\theta||_2^2$ is $L_2$ regularization, weighted by $\lambda$.

To facilitate this learning process, the feature space, denoted as $Z$, which was originally extracted by the expert policy was set to train the network. By capturing the essential characteristics of the expert's navigation strategy, this feature space serves as a meaningful representation of the observations (Hou et al., 2019). Subsequently, we train the mapping from the learned feature space $Z$ to actions, allowing the behavioral cloning algorithm to effectively mimic the expert's decision-making process. Through this iterative process of learning and mapping, our behavioral cloning algorithm learns to navigate based on the expert trajectory while using less information compared to the expert. Fig. 5 shows the concept of our imitation learning task.

**Force Prediction.** This is a crucial task in endovascular intervention, as surgeons utilize force feedback cues to avoid damaging the endothelial wall of the patient's blood vessels. Many force prediction methods have been proposed by employing sensor utilization (Yokoyama et al., 2008) or image-based methods (Song et al., 2022). We present a supervised method to demonstrate the force prediction capabilities of our ENN. The structure of our force prediction algorithm consists of a CNN coupled with an MLP head and the following loss function:

$$\mathcal{L} = \mathcal{L}_Z + \mathcal{L}_f = \sum_{i=1}^{D}(Z_i - \hat{Z}_i)^2 + \sum_{i=1}^{D}(f - \hat{f})^2 \ . \tag{3}$$

where $\hat{Z}$ represents the feature vector extracted by ENN and $\hat{f}$ represents the force resulted from the transition $\pi_{\exp}$, $D$ represents the number of samples in the collected dataset.

## 5 EXPERIMENTS

We first validate the realism of our CathSim and then analyze the effectiveness of the ENN. Since other endovascular simulators are closed-source or do not support learning algorithms, it is not straightforward to compare our CathSim with them. We instead compare our CathSim with the real robot to show that our simulator can mimic the real robot's behavior. We note that our experiments mainly focus on benchmarking CathSim and learning algorithms. Other aspects of our simulator such as blood simulation and AR/VR are designed for surgical training and, hence are not discussed in this paper. The details of the running speed of our simulator are discussed in the Appendix B.

### 5.1 CATHSIM VALIDATION

**CathSim vs. Real Robot Comparison.** To assess our simulator's accuracy, we juxtaposed the force measurements from our simulator with those from real-world experiments. In the real experiments Kundrat et al. (2021), an ATI Mini40 load cell was used to capture the force resulting from the interaction between instruments and the same Type-I silicone phantom employed in our experiments. This force-based comparison was chosen due to the scarcity of quantitative metrics suitable for evaluations (Rafii-Tari et al., 2017). The setup details are provided in our Appendix D.

**Statistical Analysis.** We compare the observed empirical distribution and a normal distribution derived from the real experiments conducted by Kundrat et al. (2021). We derive a cumulative

distribution (Fig. 6) by sampling data from a Gaussian distribution given the experiments by Kundrat et al. (2021). We utilize Mann-Whitney test to compare the given distributions. The resulting statistic given the test is 76076, with a p-value of, $p \approx 0.445$ which leads to the conclusion that the differences in the distributions are merely given to chance. As such, the distributions can be considered as being part of the same population and thus convene that the force distribution of our simulator closely represents the distribution of forces encountered in the real-life system. Therefore, we can see that our CathSim successfully mimics the behavior of the real-world robotic system.

**User Study.** We conducted a user study with 10 participants, asking them to evaluate our CathSim based on seven key criteria: 1) anatomical accuracy 2) navigational realism 3) user satisfaction 4) friction accuracy 5) interaction realism 6) motion accuracy 7) visual realism All user study questions are available in Appedix G. Table 2 shows the results, where the responses are represented in a 5-point Likert scale. Despite comprehensive positive feedback, enhancement of the simulator's visual experience was identified as an area for improvement.

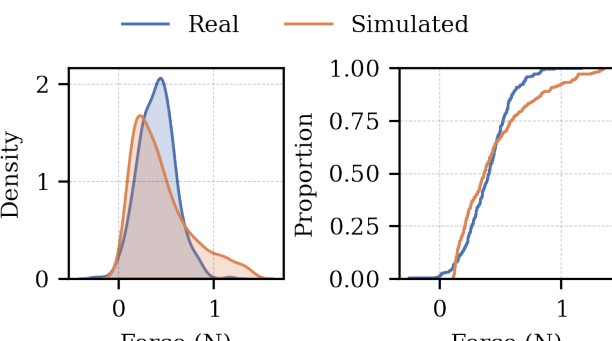

Table 2: User-study results.

| Question | Average | STD |
|---|---|---|
| Anatomical Accuracy | 4.57 | 0.53 |
| Navigation Realism | 3.86 | 0.69 |
| User Satisfaction | 4.43 | 0.53 |
| Friction Accuracy | 4.00 | 0.82 |
| Interaction Realism | 3.75 | 0.96 |
| Motion Accuracy | 4.25 | 0.50 |
| Visual Realism | 3.67 | 1.15 |

Figure 6: Comparison between the simulated force from our CathSim and real force from the real robot.

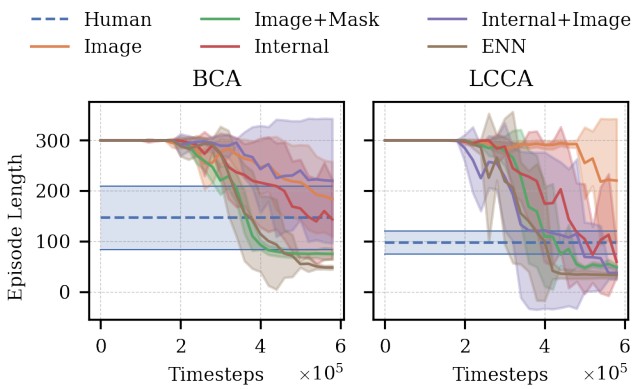

Figure 7: Episode lengths of when utilizing different input modalities.

Figure 8: Experiment Setup

Table 3: Force prediction results.

| Algorithm | MSE (N) ↓ |
|---|---|
| Baseline | 5.0021 |
| FPN (1K) | 0.5047 |
| FPN (10K) | 0.1828 |
| FPN (100K) | 0.0898 |

## 5.2 EXPERT TRAJECTORY ANALYSIS

**Experimental Setup.** We conduct the experiment to validate the effectiveness of the expert trajectory with different modality inputs (i.e., image, internal, segmentation mask). We also employ a professional endovascular surgeon who *controls CathSim manually* to collect the "Human" trajectory. We propose the following metrics (details in Appendix C) to evaluate the catheterization results: Force (N), Path Length (cm), Episode Length (steps), Safety (%), Success (%), and SPL (%). Two targets are selected for the procedures, specifically the brachiocephalic artery (BCA) and the left common carotid artery (LCCA). The targets and the initial placement of the catheter and the targets are visualized in Fig. 8. More details on experiment setups can be found in Appendix E. In

all training setups, our *CathSim's speed is from* 40 *to* 80 *frames per second*, which is well suited for real-time applications.

Table 4: Expert navigation results. ENN uses both image, internal, and segmentation mask as inputs.

| Target | Input | Force | Path Length | Episode Length | Safety | Success | SPL |
| | | (N) ↓ | (cm) ↓ | (s) ↓ | % ↑ | % ↑ | % ↑ |
| --- | --- | --- | --- | --- | --- | --- | --- |
| BCA | Human | $\mathbf{1.02 \pm 0.22}$ | $28.82 \pm 11.80$ | $146.30 \pm 62.83$ | $\mathbf{83 \pm 04}$ | $100 \pm 00$ | 62 |
| | Image | $3.61 \pm 0.61$ | $25.28 \pm 15.21$ | $162.55 \pm 106.85$ | $16 \pm 10$ | $65 \pm 48$ | 74 |
| | Image+Mask | $3.36 \pm 0.41$ | $18.55 \pm 2.91$ | $77.67 \pm 21.83$ | $25 \pm 07$ | $100 \pm 00$ | 86 |
| | Internal | $3.33 \pm 0.46$ | $20.53 \pm 4.96$ | $87.25 \pm 50.56$ | $26 \pm 09$ | $97 \pm 18$ | 80 |
| | Internal+Image | $2.53 \pm 0.57$ | $21.65 \pm 4.35$ | $221.03 \pm 113.30$ | $39 \pm 15$ | $33 \pm 47$ | 76 |
| | **ENN** | $2.33 \pm 0.18$ | $\mathbf{15.78 \pm 0.17}$ | $\mathbf{36.88 \pm 2.40}$ | $45 \pm 04$ | $100 \pm 00$ | 99 |
| LCCA | Human | $\mathbf{1.28 \pm 0.30}$ | $20.70 \pm 3.38$ | $97.36 \pm 23.01$ | $\mathbf{77 \pm 06}$ | $100 \pm 00$ | 78 |
| | Image | $4.02 \pm 0.69$ | $24.46 \pm 5.66$ | $220.30 \pm 114.17$ | $14 \pm 14$ | $33 \pm 47$ | 69 |
| | Image+Mask | $3.00 \pm 0.29$ | $16.32 \pm 2.80$ | $48.90 \pm 12.73$ | $33 \pm 06$ | $100 \pm 00$ | 96 |
| | Internal | $2.69 \pm 0.80$ | $22.47 \pm 9.49$ | $104.37 \pm 97.29$ | $39 \pm 17$ | $83 \pm 37$ | 79 |
| | Internal+Image | $2.47 \pm 0.48$ | $14.87 \pm 0.79$ | $37.80 \pm 10.50$ | $42 \pm 08$ | $100 \pm 00$ | 100 |
| | **ENN** | $2.26 \pm 0.33$ | $\mathbf{14.85 \pm 0.79}$ | $\mathbf{33.77 \pm 5.33}$ | $45 \pm 05$ | $100 \pm 00$ | 100 |

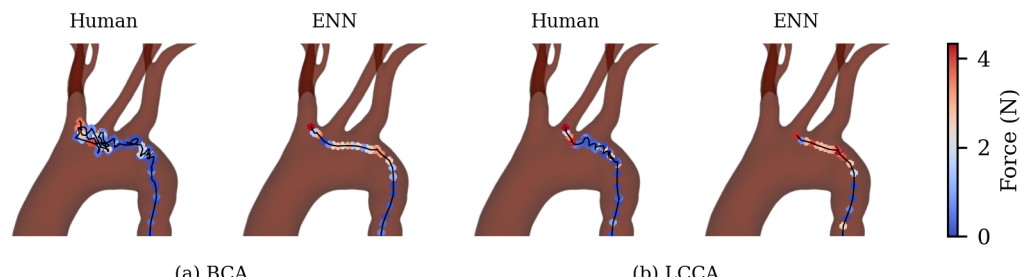

(a) BCA                                    (b) LCCA

Figure 9: Examples of navigation path from an endovascular surgeon and our ENN.

**Quantitative Results.** Table 4 shows that the expert network ENN outperforms other tested configurations for both BCA and LCCA targets. It excels in terms of minimal force exerted, the shortest path length, and the least episode length. While the human surgeon shows a better safety score, ENN surpasses most other configurations. The results show that utilizing several modality inputs effectively improves catheterization results. However, we note that the *human surgeon still performs the task more safely* in comparison with ENN, and safety is a crucial metric in real-world endovascular intervention procedures.

**Expert Trajectory vs. Humans Skill.** To evaluate the performance of various iterations of our model during training, we computed the mean episode length and compared it with the human results. As depicted in Fig. 7, within the BCA, our algorithms successfully navigate the environment after $10^5$ time steps and half of them exhibited superior performance compared to the human operator. Moreover, it is evident from both targets' navigation that ENN consistently achieves good performance with low inter-seed variance. We note although our ENN outperforms the surgeon, *ENN uses several modality inputs* (including image, internal force, segmentation mask), while the *surgeon only relies on the image* to conduct the task.

**Navigation Path.** Fig. 9 shows the comparison between the navigation paths generated by our ENN and a surgeon. The surgeon's path exhibits a meandering trajectory, which stands in contrast to the expert's more direct route. Moreover, the path taken by the human operator in navigating towards the BCA (Fig. 9(a)), demonstrates heightened irregularity, which indicates the increased difficulty in comparison to targeting the LCCA. This is likely due to the BCA's deeper location within the chest. Despite these challenges, the human operator exerted less force compared to our ENN algorithm.

Table 5: The imitation learning results with and without using the expert navigation network (ENN).

| Target | Algorithm | Force (N) ↓ | Path Length (cm) ↓ | Episode Length (s) ↓ | Safety % ↑ | Success % ↑ | SPL % ↑ |
|---|---|---|---|---|---|---|---|
| BCA | ENN | $2.33 \pm 0.18$ | $\mathbf{15.78 \pm 0.17}$ | $\mathbf{36.88 \pm 2.40}$ | $45 \pm 04$ | $100 \pm 00$ | $\mathbf{99}$ |
| | Image w/o. ENN | $3.61 \pm 0.61$ | $25.28 \pm 15.21$ | $162.55 \pm 106.85$ | $16 \pm 10$ | $65 \pm 48$ | $74$ |
| | Image w. ENN | $\mathbf{2.23 \pm 0.10}$ | $16.06 \pm 0.33$ | $43.40 \pm 1.50$ | $\mathbf{49 \pm 03}$ | $100 \pm 00$ | $98$ |
| LCCA | ENN | $\mathbf{2.26 \pm 0.33}$ | $14.85 \pm 0.79$ | $33.77 \pm 5.33$ | $\mathbf{45 \pm 05}$ | $100 \pm 00$ | $100$ |
| | Image w/o ENN | $4.02 \pm 0.69$ | $24.46 \pm 5.66$ | $220.30 \pm 114.17$ | $14 \pm 14$ | $33 \pm 47$ | $69$ |
| | Image w. ENN | $2.51 \pm 0.21$ | $\mathbf{14.71 \pm 0.20}$ | $\mathbf{33.10 \pm 2.07}$ | $43 \pm 04$ | $100 \pm 00$ | $100$ |

## 5.3 DOWNSTREAM TASK RESULTS

**Imitation Learning.** The details of our imitation learning setup can be found in the Appendix F. A comparison of the baseline algorithm (i.e., using only Image), the expert (ENN), and the behavioral cloning algorithm utilizing ENN is presented in Table 5. The expert trajectories produced by ENN have a significant impact on algorithm performance within a limited observation space. Compared to the Image baseline, utilizing expert trajectories in the behavioral cloning algorithm leads to re-markable improvements. Both for the BCA and LCCA targets, the integration of expert trajectories results in lower force, shorter path and episode length, higher safety, and a high success rate and SPL score. This demonstrates the potential of expert trajectories to enhance performance beyond the baseline, indicating their value for sim-to-real transfer in future research.

**Force Prediction.** Table 3 shows the impact of the amount of generated trajectory samples, obtained by the use of ENN, on the fine-tuning of the Force Prediction Network (FPN) and its subsequent per-formance, measured through the Mean Square Error (MSE). The baseline MSE stands at $5.0021$N. When we fine-tuned the FPN with $1,000$ (1K) generated samples, the MSE was reduced signifi-cantly to $0.5047$N, demonstrating the efficacy of the expert network in generating valuable samples for network training. As the quantity of generated samples increased to $10,000$ (10K), the MSE further dropped to $0.1828$N. This trend continued, as evidenced by the decrease in MSE to $0.0898$N when the FPN was fine-tuned with $100,000$ (100K) samples. These results highlight the potential of the expert network to generate increasingly useful samples for training which is unattainable by human participants in real-world procedures, and the subsequent ability of the FPN to be fine-tuned to achieve progressively better results.

## 6 DISCUSSION

Our work has introduced the first open-source and real-time endovascular simulator. To this end, we do not propose any new learning algorithms, instead, our CathSim serves as the foundation for future development. Similar to the autonomous driving field (Dosovitskiy et al., 2017) where the simulators significantly advance the field, we hope that our CathSim will attract more attention and work from the machine learning community to tackle the autonomous catheterization task, an important but *relatively underdeveloped research direction*.

**Limitation.** While we have demonstrated the features of CathSim and successfully trained an expert navigation network, there are notable limitations. First, since CathSim is the very first open-source endovascular simulator, it is infeasible for us to compare it with other closed-source ones. However, we have validated our simulator with a real robot setup. We hope that with our open-source simu-lator, future research can facilitate broader comparisons. Additionally, it's pertinent to note that the expert trajectory utilized in our study is generated using CathSim. This may introduce an inherent bias, as the trajectory is specific to our simulator's environment and might not fully replicate real-world scenarios. Second, in order to simplify the simulation and enable real-time factors, we utilize rigid body and rigid contact assumptions, which do not fully align with the real world where the aorta is deformable and soft. This aspect potentially limits the applicability of our findings in real-world settings. Finally, we have not applied our learned policy to the real robot as our ENN shows strong results primarily under multiple modalities input, which is not feasible in actual procedures reliant solely on X-ray images for navigation.

Although the *ENN outperforms human surgeons* in some metrics, we clarify that our results are achieved *under assumptions*. First, our ENN utilizes multiple inputs including joint positions, joint velocities, internal force, and segmentation images, while surgeons rely solely on the images. Additionally, the surgeon interacts with the simulator using traditional devices (keyboards), which may limit precision due to the discretized action space. In contrast, our algorithm operates continuously. These differences afford our ENN certain advantages, thereby contributing to its promising results.

**Future Work.** We see several interesting future research directions. First, we believe that apart from training expert navigation agents, our CathSim can be used in other tasks such as planning, shape prediction, or sim2real X-ray transfer (Kang et al., 2023). We provide initial results and the baselines of these tasks in Appendix H, I. Second, extending our simulator to handle soft contact and deformation aorta would make the simulation more realistic. Third, our CathSim simulator can be used to collect and label data for learning tasks. This helps reduce the cost, as real-world data collection for endovascular intervention is expensive (Kundrat et al., 2021). Finally, developing robust autonomous catheterization agents and applying them to the real robot remains a significant challenge. Currently, it is uncertain how seamlessly the learning agent's policy would adapt to real-world situations under real-world settings such as dynamic blood flood pressure, soft and deformable tissues. We believe several future works are needed to improve both the endovascular simulators and learning policy to bridge the simulation to real gap.

Finally, we note that our CathSim is a simulator for medical-related tasks. Therefore, agents trained in our environment must not be used directly on humans. Users who wish to perform real-world trials from our simulator must ensure that they obtain all necessary ethical approvals and follow all local, national, and international regulations.

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

# A CATHSIM DESIGN DETAILS

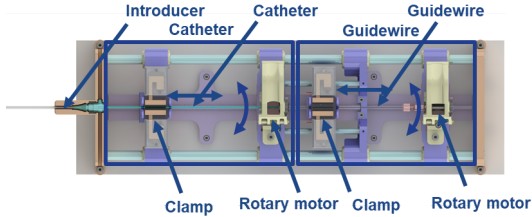 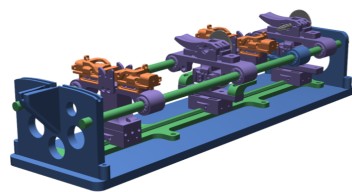

(a) The design of the Robotic Follower (Slave Robot)  (b) Our simulated version of the Robotic Follower

Figure 10: Schematic representation of the CathBot's follower mechanism (Kundrat et al., 2021) (a) alongside a visualization of our simulated model (b).

**Robotic Follower.** In our study, we focus on simulating the robotic follower, predicated on the linear relationship between the leader and follower, as outlined in the CathBot design (Kundrat et al., 2021). For the sake of simplicity, our simulation comprises four modular platforms attached to the main rail; two of these platforms hold the guidewire during translational movements with clamps, while the other two facilitate angular movements via rotary catheter and guidewire platforms. Prismatic joints connect the main rail components and the clamps, enabling translational movements, while revolute joints link the wheels, allowing the catheter and guidewire to rotate (refer to Fig. 10 for the design).

**Actuation.** CathBot's actuation entirely depends on the frictional forces $f_f$ between the guidewire and the clamp. In our simplified model, we assert that the frictional force $f_f$ is sufficient to entirely prevent slippage ($f_f \geq f_s$), hence eliminating the need to account for the effects of sliding friction $f_s$. This approach gives us direct control over the joints without having to simulate frictional effects, leading to faster simulation times due to fewer contact points. Furthermore, a friction-based actuation mechanism could potentially slow down execution times and increase error probability due to simulation noise. Thus, we argue that our choice to assume perfect motion results in enhanced computational efficiency, particularly within the context of our defined problem domain.

**Aortic Models.** In addition to Type-I Aortic Arch model which is mainly used in our experiments, we incorporate three distinct aortic models to enrich our anatomical dataset. These models include a high-fidelity Type-II aortic arch and a Type-I aortic arch with an aneurysm, both sourced from Elastrat, Switzerland. Furthermore, a low tortuosity aorta model, based on a patient-specific CT scan, is included. With these three additional representations, our simulator contains four distinct aorta models. These models aim to enhance the diversity and accuracy of aortic structures available for research and educational endeavors. These aortas are visualized in Fig. 11.

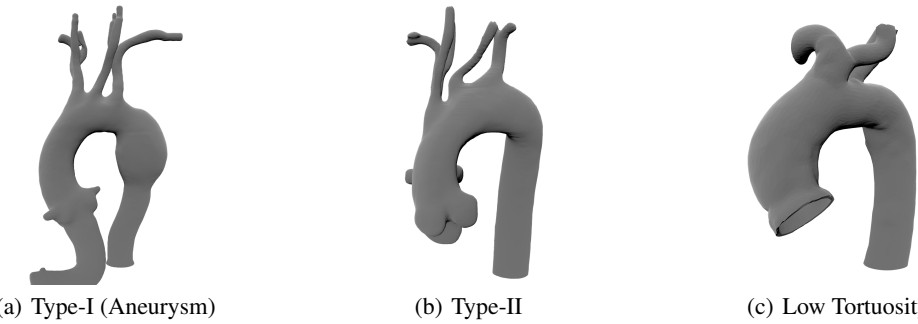

(a) Type-I (Aneurysm)          (b) Type-II          (c) Low Tortuosity

Figure 11: Aortic Models.

## B CATHSIM SPEED EVALUATION

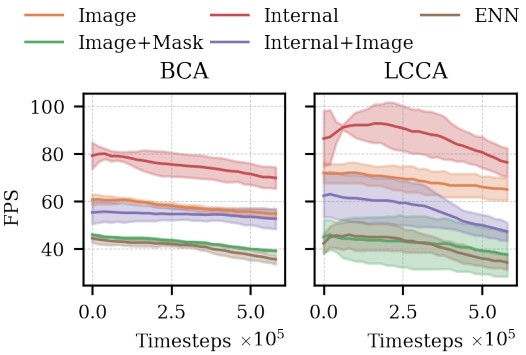

Table 6: Comparative training times

| Algorithm | Training Time (h) | |
|---|---|---|
| | BCA | LCCA |
| Image | $3.00 \pm 0.11$ | $2.54 \pm 0.17$ |
| Image+Mask | $4.20 \pm 0.05$ | $4.60 \pm 1.29$ |
| Internal | $2.38 \pm 0.15$ | $2.20 \pm 0.18$ |
| Internal+Image | $3.15 \pm 0.28$ | $3.54 \pm 0.29$ |
| **ENN** | $4.61 \pm 0.22$ | $4.83 \pm 0.41$ |

Figure 12: CathSim training speed.

**Training Speed.** As illustrated in Fig. 12, we provide a comparison of frames per second (FPS) for the various algorithms we employed during model training. It is evident that utilizing solely the internal state space, comprised of joint positions and velocities, facilitates expedited training processes. In contrast, integrating all modalities into the training process results in its deceleration. The most significant computational demand arises from the dual convolutional neural networks utilized in both the image and mask representations. However, despite this load, the algorithms exhibit respectable computational speed, even during the training phase. Our simulator supports approximately 40 to 80 frames per second performance for all implemented algorithms, underscoring the computational speed of our simulation environment. Moreover, we provide the training times in terms of hours for the different modalities in Table 6.

## C EVALUATION METRICS

**Force.** In our study, the force applied by surgical instruments during the simulation is critical for evaluating their performance and interaction with the aorta. To accurately record this force, we used a manual cannulation method through our simulator. At each time step of the simulation, we collected the collision points between the guidewire and the aorta. These collision points give us crucial insights into the tridimensional force acting on the system, which consists of the normal force ($f_z$) and the frictional forces ($f_x$ and $f_y$). To compute the total magnitude of the force, we calculated the Euclidean norm of the force vector at a given time step ($t$), denoted as $f_t$. This magnitude is obtained from the square root of the sum of the squares of the force vector components

$$f_t = \sqrt{f_{x,t}^2 + f_{y,t}^2 + f_{z,t}^2} \tag{4}$$

This allows us to holistically assess the collective impact of the force components, providing a more comprehensive understanding of the guidewire's behavior and its interaction forces with the aorta throughout the simulation. Furthermore, it facilitates a comparison between our experiments and those conducted by Kundrat et al. (2021).

**Path Length.** The path length was derived by summing the Euclidean distances between sequential positions of the guidewire head. For each time step, the position of the guidewire head, denoted as $h_t$, was extracted. The Euclidean distance between the guidewire head position at time $t$ and the position at time $t + 1$, denoted as $h_t$ and $h_{t+1}$ respectively, was then calculated $d(h_t, h_{t+1}) = \sqrt{(h_{t+1} - h_t)^2}$. This process resulted in a path length represented by:

$$\text{PathLength} = \sum_{i=1}^{n} ||h_{t+1} - h_t|| \tag{5}$$

where $n$ is the episode length.

**SPL.** The navigation performance of the expert was evaluated in relation to human performance. This involved utilizing the path length, as calculated in the previous paragraph, to assess the optimality of the navigation. An optimal policy results in the shortest path. The Success Weighted by Normalized Inverse Path Length (SPL) metric, as suggested by Anderson et al. (2018) was then computed using

$$\text{SPL} = \frac{1}{N} \sum_{i=1}^{N} S_i \frac{l_i}{\max(p_i, l_i)} \tag{6}$$

where the path length $p_i$ is normalized by the optimal path $l_i$. In this context, the shortest path observed was used as the optimal path, considering that human performance is consistently outperformed by the RL policies.

**Safety.** We compute the safety based on the number of times an algorithm inflicts a force greater than 2N. This constant is derived from the study in real-world setup (Kundrat et al., 2021). As such, we create a binary variable $a \in \{0, 1\}$ where $a = 1$ if $f_t \geq 2$N. As such, the safety metric is defined as:

$$\text{Safety} = 1 - \frac{1}{N} \sum_{i=1}^{N} a_i \tag{7}$$

where $N$ represents the number of steps within an episode. We further subtract the result from $1$. Intuitively, an algorithm that inflicts a great force at each step during the episode will have a safety of $0\%$, whereas an algorithm that inflicts a damage of $f \leq 2$N at all time steps, will result in a safety of $100\%$.

**Episode Length.** The length of an episode is determined by the number of steps an algorithm takes to complete a task. This metric is significant as it provides insight into the efficiency of an algorithm; fewer steps generally indicate more efficient performance, assuming that the quality of task completion is preserved.

**Success.** The success of an episode is defined by whether the agent is able to achieve the goal within a pre-specified time limit of $300$ time steps. This metric is binary; it records a success if the goal is reached within the time limit, and a failure otherwise. This serves to measure the effectiveness of the agent in task completion under time constraints, mirroring real-world scenarios where timeliness is often crucial.

## D  CathSim Validation Against Real-Robot

**Force Extraction.** The real robot experiments conducted by Kundrat et al. (2021) employed a 6 DoF force sensor (Mini40, ATI Industrial Automation, Apex, USA) to measure the force produced by the interaction between the instruments and the silicone phantom (see Fig. 13). The force sensor was placed directly underneath the phantom, where the magnitude of the force was extracted. Similarly, in order to extract the force within our simulator, we manually carry out the cannulation using an input device and extract the forces at each time step. For each simulation time step, we extract the collision points between the guidewire and the aorta, along with the tridimensional force expressed as the normal force $f_z$ and frictional forces $f_x$ and $f_y$. We then compute the force magnitude using Equation 4.

**Distribution Comparison.** We begin our comparison with the Shapiro-Wilk test of normality on the data extracted from the simulator and, given a p-value of $p \approx 7.195 \times 10^{-17}$ and a statistic of $0.878$, we conclude that the sampled data does not represent a normal distribution $\mathcal{N}(\mu, \sigma)$. Furthermore, we assess the homoscedasticity of the sample distribution and normal distribution by using Levene's tests, which results in a statistic of $40.818$ and a p-value of $p \approx 2.898 \times 10^{-10}$, therefore, concluding that $\sigma_1^2 \neq \sigma_2^2$. Given the previous statistics (i.e., non-normal distribution and unequal variances), we select the non-parametric Mann-Whitney test to compare the given distributions. The statistical tests show that our CathSim can mimic the real-robot behavior, as indicated in the main paper.

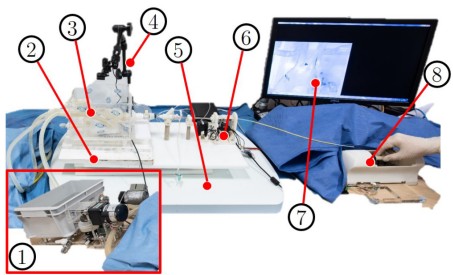

Figure 13: Experimental setup of CathBot: (1) Pulsatile and continuous flow pumps, (2) Force Sensor, (3) Vascular phantom, (4) Webcam, (5) NDI Aurora field generator, (6) Catheter manipulator (i.e., robotic follower), (7) Simulated X-ray Screening, and (8) Master device. Adapted from Chi et al. (2020); Kundrat et al. (2021).

## E EXPERT NAVIGATION NETWORK IMPLEMENTATION

**Observation Space.** We incorporated a range of observations to provide the agent with an extensive understanding of the environment. A grayscale image of dimensions $80 \times 80$ (denoted by $I \in [0,1]^{80 \times 80}$) is generated as a visual clue, which is accompanied by the ground truth binary segmentation mask $S \in \{0,1\}^{80 \times 80}$. The mask $S[i,j] = 1$ is validated only when the pixel coordinates $(i,j)$ are part of set $A$, the pixels that constitute the guidewire. Furthermore, to enhance the reinforcement learning problem's optimization, we also included the ground truth joint positions $Q \in \mathbb{R}^{168}$ and joint velocities $V \in \mathbb{R}^{168}$. These joint values provide detailed mechanical insight into the guidewire's state, enriching the agent's knowledge base and facilitating informed decision-making.

**Action Representation.** In CathSim, the essential actions are denoted by translation $a_t \in [-1,1]$ and rotation $a_r \in [-1,1]$. At each time step, the agent generates these actions, which are collectively represented by the vector $a \in [-1,1]^2$. Here, a positive $a_t$ denotes forward movement, while a negative $a_t$ signifies a backward movement. Similarly, a positive $a_r$ corresponds to a clockwise guidewire rotation, while a negative $a_r$ indicates an anticlockwise rotation. The selection of translation and rotation actions in CathSim closely mirrors the actions in real-world endovascular procedures, as described by Kundrat et al. (2021). By faithfully reproducing the motions of an actual robot, the simulation environment better replicates real-world situations. This enables the reinforcement learning agent to acquire policies that can be feasibly implemented in a physical robot, thereby augmenting the practical value and applicability of the behaviors learned.

**Reward Function.** Our employed reward system is dense, and it hinges on the spatial position of the guidewire tip in relation to the goal. Formally, the reward $r$ is determined by the function $r(p_t, g) = -d(p_t, g)$, where $d(p_t, g) = ||p_t - g||$ characterizes the distance between the position $p$ of the guidewire tip at time $t$ and the target $g$. If the guidewire tip lies within a distance threshold $\delta$ of the target $g$, the agent is conferred a positive reward $r$. For our research, we adopted a $\delta$ of 4mm and assigned a task completion reward of $r = 10$. Consequently, we end up with the following reward function:

$$r(h_t, g) = \begin{cases} 10 & \text{if} \quad d(h, g) \leq \delta \\ -d(h, g) & \text{otherwise} \end{cases} \tag{8}$$

**Training Details.** The experiments were conducted on an NVIDIA RTX 2060 GPU (33MHz) system on an Ubuntu 22.04 LTS based operating system. Furthermore, the system contained an AMD Ryzen 7 5800X 8-Core Processor with a total of 16 threads with 64GB of RAM. All experiments used PyTorch, whilst for the Soft Actor Critic implementation we used stable baselines (Raffin et al., 2021). The training was carried out for a total of $600,000$ time steps using a total of 5 different random seeds, resulting in a training time bounded between 2 and 5 hours. Each episode has two terminal states, one which is time-bound (i.e., termination of an episode upon reaching a number of steps) and one which is goal bound (i.e., the agent achieves the goal $g$).

Table 7: The network architectures for ENN.

| Network | Layer (type) | Output Shape | Param # | Nonlinearity |
|---------|--------------|--------------|---------|--------------|
| CNN | Input | (1, 80, 80) | 0 | - |
| | Conv2D | (32, 19, 19) | 2080 | ReLU |
| | Conv2D | (64, 8, 8) | 32832 | ReLU |
| | Conv2D | (64, 6, 6) | 36928 | ReLU |
| | Flatten | (2304) | 0 | - |
| | Linear | (256) | 590080 | ReLU |
| MLP | Input | (1, 336) | 0 | - |
| | Linear | (256) | 86272 | ReLU |
| | Linear | (128) | 32896 | ReLU |

Table 8: SAC hyperparameters

| Hyperparameter | Value |
|----------------|-------|
| Learning Rate | $3 \times 10^{-4}$ |
| Buffer Size | $10^6$ |
| Batch Size | 256 |
| Smoothing Coefficient ($\tau$) | 0.005 |
| Discount ($\gamma$) | 0.99 |
| Train Frequency | 1 |
| Gradient Steps | 1 |
| Entropy Coefficient | 1 |
| Target Update Interval | 1 |
| Target Entropy | -2 |

**Networks.** We employ multiple feature extractors to dissociate the dominant features within our Expert Navigation Network (ENN). Specifically, we use a convolutional neural network (CNN (Mnih et al., 2015)) to extract the image-based features, resulting in two latent features $J_I$ and $J_S$ that represent the top camera view and guidewire segmentation map. The CNN is composed of 3 convolutional layers with a ReLU (Agarap, 2018) activation function, followed by a flattening operation. We also concatenate joint positions $Q$ and joint velocities $V$ to generate a joint feature vector $J_J = Q \parallel P$ of dimensionality 336 which is then passed through the MLP. These features are concatenated to form a single feature vector $J = J_I \parallel J_S \parallel J_J$, which is fed into a policy network $\pi(a_t, \theta)$. Both network architectures are presented in Table 7.

**SAC.** Our primary reinforcement learning method is the soft actor-critic (SAC) (Haarnoja et al., 2018). Soft actor-critic (SAC) is a model-free reinforcement learning algorithm that learns a stochastic policy and a value function simultaneously. The objective function of SAC combines the expected return of the policy and the entropy of the policy, which encourages exploration and prevents premature convergence to suboptimal policies. The algorithm consists of three networks, namely a state-value function $V$ parameterized by $\psi$, a soft $Q$-function $Q$ parameterized by $\theta$, and a policy $\pi$ parameterized by $\phi$. The parameters we employ are present in Table 8. For the policy network, we use a composition of linear layers to handle and transform the input data.

## F DOWNSTREAM TASKS

### F.1 IMITATION LEARNING

**Network.** The architecture of our network for the Behavioral Cloning (BC) algorithm consists of a sequence of specialized layers that gradually transform the input data. The process initiates with the input, formatted as a tensor of dimensions (1, 80, 80), which is passed through three Conv2D layers. All of these layers employ the Rectified Linear Unit (ReLU) as their activation function. Upon passing through the Conv2D layers, the output tensor is reshaped into a single dimension by a flattening layer, facilitating the shift from convolutional to linear layers. The flattened output is then processed by a Linear layer utilizing the ReLU activation function. Finally, an additional Linear layer, without an activation function, delivers the final output of the network. This last layer provides raw scores, which are interpreted directly as the network's output, mirroring the decisions made by the BC algorithm. The network architecture is illustrated in Table 9.

Table 9: BC Architecture

| Layer (type) | Output Shape | Param # | Nonlinearity |
|--------------|--------------|---------|--------------|
| Input | (1, 80, 80) | 0 | - |
| Conv2D | (32, 19, 19) | 2080 | ReLU |
| Conv2D | (64, 8, 8) | 32832 | ReLU |
| Conv2D | (64, 6, 6) | 36928 | ReLU |
| Flatten | (2304) | 0 | - |
| Linear | (512) | 1180160 | ReLU |
| Linear | (2) | 1026 | None |

Table 10: BC hyperparameters

| Hyperparameter | Default Value |
|----------------|---------------|
| Batch Size | 32 |
| Optimizer | Adam |
| Learning Rate | $1 \times 10^{-3}$ |
| Entropy Coefficient | $1 \times 10^{-3}$ |
| Epochs | 800 |

**Training.** To train the BC algorithm, we employ a set of hyperparameters, as listed in Table 10. The training procedure makes use of a batch size of 32. The Adam (Kingma & Ba, 2014) optimizer is

utilized to minimize the loss function. We set the learning rate and entropy coefficient to $1 \times 10^{-3}$, controlling the step size at each iteration of the optimization and regularizing the policy towards a more exploratory behavior, respectively. The training process spans over $800$ epochs, each epoch consisting of a complete pass through the entire dataset, providing the model ample opportunity to converge to an optimal solution.

## F.2 FORCE PREDICTION

Table 11: Force Prediction Network Architecture

| Layer (type) | Output Shape | Param # | Nonlinearity |
|---|---|---|---|
| Input | (1, 80, 80) | 0 | - |
| Conv2D | (32, 19, 19) | 2080 | SELU |
| Conv2D | (64, 8, 8) | 32832 | SELU |
| Conv2D | (64, 6, 6) | 36928 | SELU |
| Flatten | (2304) | 0 | - |
| Linear | (768) | 1769472 | SELU |
| Linear | (256) | 196864 | SELU |
| Linear | (1) | 257 | ReLU |

**Network.** We utilize a hybrid architecture comprising a CNN followed by an MLP, with a grayscale image as the input. The network architecture includes three Conv2D layers that apply convolution operations to the input tensor. Each of these convolutional layers employs the Scaled Exponential Linear Unit (SELU (Klambauer et al., 2017)) as the activation function, encouraging self-normalization and improving the network's capability to propagate gradients deeply into the architecture. Subsequent to the Conv2D layers, a flattening layer reshapes the output into a one-dimensional tensor. Thereafter, the architecture implements three Linear layers. The initial two layers use the SELU activation function, while the final one adopts the Rectified Linear Unit (ReLU (Agarap, 2018)) activation function. This amalgamation of multiple linear layers, each paired with a distinct non-linear activation function, enables the network to learn complex patterns in the input data, leading to the network's final output. Table 11 shows the force prediction network architecture.

**Training.** We employed the NAdam optimizer (Dozat, 2016) to train our model, setting the initial learning rate at $4 \times 10^{-4}$. This was coupled with a SELU activation function to enhance self-normalization and facilitate gradient propagation. To dynamically adjust the learning rate during the training process, we used a OneCycle learning rate scheduler (Smith & Topin, 2018), and the training was conducted for a total of 30 epochs.

**Data Management.** In terms of data management, we divided the original dataset into training, testing, and validation subsets according to the total time steps. Initially, we allocated $20\%$ of the total time steps to the test set. Subsequently, the remaining data was partitioned into a validation set and training set, with the validation set representing $20\%$ of the leftover steps. This procedure was followed for each experimental run of our force prediction model, generating corresponding sets of transitions. Of particular note, the test set consistently stemmed from our largest trajectory dataset—the $100K$ dataset. For each training run, we designated $20\%$ of this dataset for validation, utilizing the remaining portion for training. The model's performance was assessed using the originally separated test set. By adhering to this structured and consistent methodology, we ensured a rigorous and equitable evaluation of our model's performance across multiple runs.

## G QUESTIONNAIRE

The study was conducted with ten volunteer participants to evaluate the effectiveness and authenticity of our endovascular simulator. These participants, who had no prior experience in endovascular navigation, provided feedback using a 5-point Likert scale (Likert, 1932). Initially, participants were shown an actual endovascular navigation fluoroscopic video. Following this, they directly interacted with the simulator, tasked with cannulating two targets: the brachiocephalic artery and the left common carotid artery. Upon task completion, participants filled out a questionnaire featuring the following prompts:

1. How well did the simulator simulate the anatomy and structure of blood vessels?
2. How closely did the simulator replicate the visual experience of navigating a real endovascular procedure?
3. How satisfied were you with the overall performance and functionality of the simulator?
4. How well did the simulation depict the resistance and friction of the guidewire against the walls of the vessel?
5. How realistic were the visual representations of the guidewire's interaction with the vessel walls?
6. How closely did the guidewire's motion match your expectations of how a real guidewire would move?
7. How visually realistic was the guidewire simulation?

## H  TRADITIONAL PATH PLANNING

To emphasize our choice of using a neural network policy as the expert, we conducted a defining experiment. We initially used the A* planner (Hart et al., 1968) to identify the most straightforward route from a starting position to the goal, ensuring sampling was restricted to the aortic arch. This plan is depicted in Fig. 14. However, the results revealed that the shortest path is not always the best. The cannulation efficiency along this route was suboptimal, and there was no standardized method to determine the exact actuation for the guidewire tip to follow this path. Expanding our research, we utilized a Multilayer Perceptron (MLP) to predict the next action from a sequence of observations. Yet, this model also faced difficulties in following the desired route, highlighting the intrinsic complexities of the task. The agent's actual path, showing the guidewire's deviation, can be seen below:

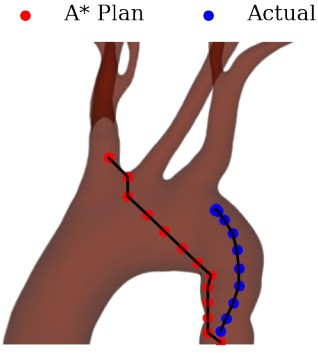

Figure 14: Path Planning

## I  SHAPE PREDICTION

Shape prediction, especially for guidewires, is instrumental in reducing vasculature stress and optimizing control performance. This ultimately results in a substantial reduction in complication risks (Shi et al., 2016). Efforts to attain precise shape prediction have led to significant investments in shape-sensing technologies. One noteworthy example is the FBG-sensors (Chitalia et al., 2020), tailored for shape approximation. However, the incorporation of these sensors presents several challenges. They not only entail material costs and augment the instrument size but also limit navigational abilities and necessitate major alterations to optimize guidewire features, including stiffness. To navigate these environments, alternatives have been sought, including reconstructions from bi-planar scanner images or those anchored in precise kinematics (Shi et al., 2016). While deep learning emerges as a promising solution, especially in terms of cost mitigation, it inherently demands extensive data.

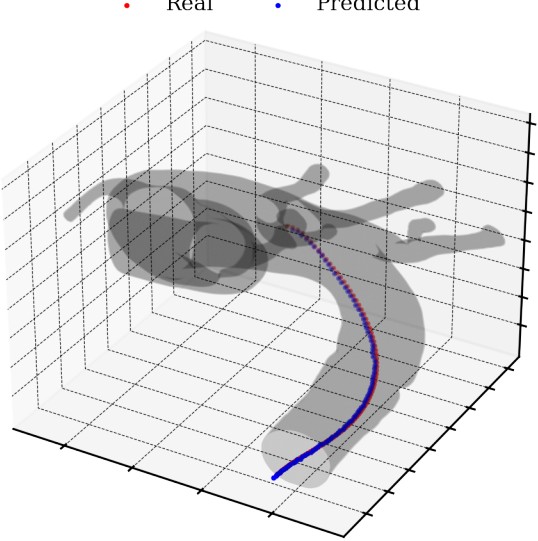

Figure 15: Shape Prediction

We posit that our simulator can be instrumental in furnishing the necessary data to train models that use images to approximate instrument shapes. To affirm this, we have juxtaposed image observations of shape $80 \times 80$ with the real 3D positions of the guidewire bodies $p$, facilitating the guidewire shape reconstruction from 2D images. Using this data, we trained a CNN, leveraging the NAdam optimizer (Dozat, 2016) and a refined loss function encompassing the Huber Loss (Huber, 1992) with a $\delta = 1$:

$$\mathcal{L}_{\text{Huber}}(y, \hat{y}) = \begin{cases} \frac{1}{2}(y - \hat{y})^2 & \text{if } |y - \hat{y}| < \delta \\ \delta\left(|y - \hat{y}| - \frac{\delta}{2}\right) & \text{, otherwise} \end{cases}, \tag{9}$$

Here, $y$ denotes the ground truth positions of the guidewire bodies, and $\hat{y}$ signifies the predicted positions. A regularization term, $\mathcal{L}_{\text{reg}}$, is introduced to ensure the guidewire's geometric structure remains intact, assuming an interbody distance of 2mm:

$$\mathcal{L}_{\text{reg}} = \frac{1}{n-1} \sum_{i=1}^{n-1} |\|\hat{y}_{i+1} - \hat{y}_i\|_2 - 0.002| \tag{10}$$

The combined loss is then represented as:

$$\mathcal{L} = \alpha \mathcal{L}_{\text{Huber}}(y, \hat{y}) + \beta \mathcal{L}_{\text{reg}}, \tag{11}$$

Our comparative analysis between the actual guidewire shape and its prediction is visually represented in Fig. 15. The close approximation of the guidewire shape from the 2D image underscores our simulator's potential.

Such findings highlight the capacity of our simulator to enhance auxiliary tasks, enabling researchers to utilize the simulator for data-driven model samples. While this provides a foundation, a deeper delve into simulation to real transfer nuances is warranted in future research. Nevertheless, considering the exorbitant costs of real-world data acquisition, our simulator stands out as a pivotal tool in the evolution of these methodologies.

## J  NETWORK CHOICE

In our experimental trials, we implemented a transformer-based architecture, drawing inspiration from the visual transformer by Dosovitskiy et al. (2020). This architecture was specifically tailored with an image size of $80$, a patch size of $10$, and a single-channel configuration, alongside a depth of $3$ and $4$ heads within the attention layers. Despite these specifications, the transformer model did not achieve the expected performance level in our specialized domain of endovascular navigation. It notably struggled to complete the designated tasks within the desired timeframe. This underperformance, coupled with signs of convergence, is illustrated in Fig 16, highlighting the challenges encountered in applying this model to our specific context.

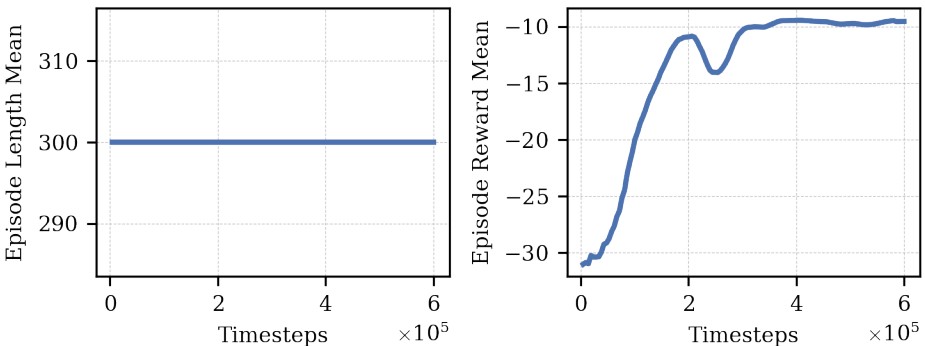

Figure 16: **ViT-Based ENN Trials:** Initial results indicate that using ViT as a feature extractor does not improve outcomes and fails to meet the target within the time limit, despite showing convergence.

## K  SIM-TO-REAL IMAGING GAP

We showcase the adaptability of our endovascular simulator using domain adaptation techniques. This is crucial for translating simulated environments into more realistic, X-ray-like images. The implementation of this approach is based on the work of Kang et al. (2023). Their method, employing multi-scale semantic matching, effectively ensures the preservation of essential structural information in medical images while transitioning from simulated to real-world imaging styles. This technique has demonstrated significant success in producing realistic X-ray images and has set a new benchmark in the field. For an in-depth understanding and visual representation of this adaptation process, we refer readers to the detailed findings and illustrations in the work of Kang et al. (2023). The adapted images can be visualized in Fig. 17.

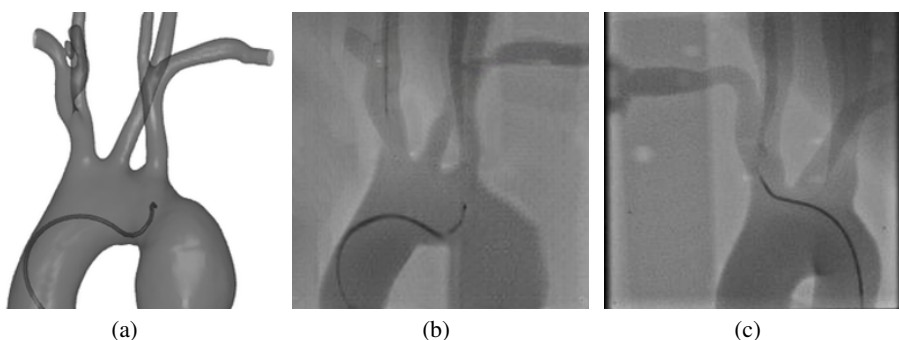

(a)  (b)  (c)

Figure 17: **Sim-to-Real Adaptation:** Demonstration of sim-to-real domain adaptation for creating realistic renderings, featuring *a)* the input image, *b)* the resultant generated image, and *c)* a real X-ray image example. Figure addapted from Kang et al. (2023)

