# OpenReview forum: "Autonomous Catheterization with Open-source Simulator and Expert Trajectory"
_ICLR.cc/2024/Conference — Submitted to ICLR 2024_

### Official Review · Reviewer_eFcL · 2023-10-27

**Soundness:** 3 good
**Presentation:** 3 good
**Contribution:** 2 fair
**Rating:** 5
**Confidence:** 3

**Summary:**

The authors develop CathSim, an open source simulator for endovascular intervention in the field of autonomous catheterization. The simulator is real-time ready, exhibits force feedback, supports Unity, and allows ML model training. The authors validate the simulator against a real robot. Furthermore, the authors use CathSim to train a multimodal expert navigation network (ENN) and show that it outperforms a human baseline on relevant metrics.

**Strengths:**

* Open source simulator
* Simulator is validated via comparison to real world robot
* Expert navigation results beating the human baseline

**Weaknesses:**

* ENN model architecture is not state-of-the-art, e.g. no transformer
* Unclear how trustworth the user study is. 10 participants in user study is a small sample size and it’s unclear how they were recruited and how trustworthy their judgements are.
* Lack of relevant details, e.g. what is the expert policy that the ENN is trained with. Why don’t you show results for the expert policy in the table?
* If an expert policy exists, why do you train an ML model (ENN) model for the expert navigation task?
* Some figures / tables are not easy to understand on their own, e.g. Table 2, for which e.g. instead of Q1 you could write “Anatomical accuracy”.

**Questions:**

Overall, this seems to be very relevant work. However, in terms of ML it is not pushing the boundary of the state-of-the-art. This work might fit better to a more applied conference.

---

> ### Author Response · Authors · 2023-11-20
> **Reviewer eFcL Response**
>
> #### Weaknesses:
>
> ##### Comment 1
>
> ENN model architecture is not state-of-the-art, e.g. no transformer
>
> **Response:**
>
> We acknowledge the absence of a transformer in the ENN model architecture. Our initial trials included experimenting with a transformer architecture, specifically based on the visual transformer framework [^1]. However, this approach did not yield the desired performance. In our experiments, the transformer-based model required approximately 4.5 hours of training but failed to successfully complete the task within the allocated time frame. Therefore, we opted for the current ENN model architecture, which demonstrated superior performance and efficiency in achieving the task objectives. We now include these details in *Appendix J*.
>
> [^1]: Dosovitskiy, A., Beyer, L., Kolesnikov, A., Weissenborn, D., Zhai, X., Unterthiner, T., Dehghani, M., Minderer, M., Heigold, G., Gelly, S. and Uszkoreit, J., 2020. An image is worth 16x16 words: Transformers for image recognition at scale. arXiv preprint arXiv:2010.11929.
>
> ---
>
> ##### Comment 2
>
> Unclear how trustworth the user study is. 10 participants in user study is a small sample size, and it’s unclear how they were recruited and how trustworthy their judgements are.
>
> **Response:**
>
> The user study involved ten student volunteers as participants. We have included this information, along with details on their recruitment and the study procedures, in *Appendix G*. Additionally, the *Discussion* section has been updated to acknowledge the non-expert status of these participants. This addresses the concerns regarding the sample size and the trustworthiness of their judgements.
>
> > *The study was conducted with ten volunteer participants to evaluate the effectiveness and authenticity of our endovascular simulator. These participants, who had no prior experience in endovascular navigation, provided feedback using a 5-point Likert scale.*
>
> ---
>
> ##### Comment 3
>
> Lack of relevant details, e.g. what is the expert policy that the ENN is trained with. Why don’t you show results for the expert policy in the table?
>
> **Response:**
>
> To clarify, the ENN in our study is a standalone system, representing a multi-modal network combined with a soft actor-critic approach. It does not utilize or train with a separate *expert policy*, hence there are no additional policy results to present in the table.
>
> ---
>
> ##### Comment 4
>
> If an expert policy exists, why do you train an ML model (ENN) model for the expert navigation task?
>
> **Response:**
>
> As per *Comment 3*, the policy is part of the network.
>
> ---
>
> ##### Comment 5
>
> Some figures / tables are not easy to understand on their own, e.g. Table 2, for which e.g. instead of Q1 you could write “Anatomical accuracy”.
>
> **Response:**
>
> Thank you for your feedback on the clarity of our figures and tables. As per your suggestion, we explicitly state the assessed properties in *Table 2* and rewrote the description to match within *Section 5.1*, under the *User Study* paragraph.
>
> | Question | Average | STD |
> |----------|---------|-----|
> | Anatomical Accuracy | 4\.57 | 0\.53 |
> | Navigation Realism | 3\.86 | 0\.69 |
> | User Satisfaction | 4\.43 | 0\.53 |
> | Friction Accuracy | 4\.00 | 0\.82 |
> | Interaction Realism | 3\.75 | 0\.96 |
> | Motion Accuracy | 4\.25 | 0\.50 |
> | Visual Realism | 3\.67 | 1\.15 |

---

> > ### Author Response · Authors · 2023-11-20
> > **Reviewer eFcL Response (continued)**
> >
> > #### Questions:
> >
> > ##### Comment 6
> >
> > Overall, this seems to be very relevant work. However, in terms of ML it is not pushing the boundary of the state-of-the-art. This work might fit better to a more applied conference.
> >
> > **Response:**
> >
> > We appreciate the feedback regarding CathSim's relevance to the ML community. To address this:
> >
> > 1. **Filling a Research Gap**: CathSim uniquely addresses the critical gap in autonomous catheterization research by offering an open-source alternative to closed-source simulators. This enhances accessibility and innovation in a field where such resources were previously limited.
> > 2. **Versatility in ML Techniques**: CathSim is not just a tool for specific applications; it supports various ML paradigms including reinforcement learning, supervised and unsupervised learning. This versatility makes it invaluable for a broad spectrum of research within the ML community.
> > 3. **Real-Time Performance**: The real-time performance of CathSim is essential for the rapid development and testing of ML algorithms, ensuring quick iteration and progress in research projects, which is a key need in the dynamic field of ML.
> > 4. **Encouraging Collaborative Research**: By offering CathSim as an open-source platform, we are not only accelerating research in autonomous catheterization, but also encouraging collaborative efforts and innovation across the ML community. This aligns with the growing trend of open science and shared progress in the field.
> >
> > In summary, CathSim's open-source nature, support for multiple ML techniques, real-time capabilities, and potential for fostering collaborative research make it a valuable asset to the ML community, particularly in advancing autonomous catheterization research.

---

> > > ### Comment · Reviewer_eFcL · 2023-11-21
> > > **Thank you for your response**
> > >
> > > Thank you for your response and the various updates to your work, which have made your publication easier to follow and understand. I also appreciate that you have added significant information in the introduction around this work's contributions to the ML community. I have updated my rating based on the updated submission.

---

> > > > ### Author Response · Authors · 2023-11-21
> > > > **Thank you**
> > > >
> > > > Dear Reviewer **eFcL**,
> > > > Many thanks for your comments. While our paper hasn't passed the acceptance bar based on your judgment, we very much appreciate your feedback and comments which definitely help improve our paper.
> > > > Thank you.

---

### Official Review · Reviewer_HLTP · 2023-10-30

**Soundness:** 3 good
**Presentation:** 3 good
**Contribution:** 3 good
**Rating:** 5
**Confidence:** 4

**Summary:**

The paper introduces an open-source simulator, CathSim, designed to generate real-time outputs to facilitate autonomous catheterization.  CathSim is used to generate semantic segmentations of guidewires, joint position, joint velocity, top camera images that are fed into a learning algorithm (CNN+MLP) to provide inputs into a reinforcement learning algorithm (soft actor critic) to predict desired catheter positions (this system is referred to as expert navigation network (ENN)).
ENN is evaluated on whether it can learn to imitate expert trajectories (imitation learning) and predict force. Various quantitative and qualitative results are provided:

* Authors demonstrate that force distributions between real robots and CatSim simulation are nearly identically using hypothesis testing (p-value analysis).
* User study assessing CatSim realism and user satisfaction is provided.
* Evaluation of trajectory planning, where two trajectories are compared: the first is generated by   a human expert using CatSim and the second is generated using ENN (automatic). Results show that ENN is competitive to human, using less force, shortest path, least episode length. Human output is safer. Also, ENN can use multimodal inputs (image, joints) while human relies only on images to perform trajectory planning. An ablation study with type of inputs is included.
* Evaluation of whether ENN can imitate the human annotated trajectory (with respect to five considered metrics, Table 5) with an ablation study.

**Strengths:**

* The paper presents an innovative idea of generating simulated outputs for improving catheter path planning that generated data for learning path planning.  A network trained using this data outperforms human surgeon (to an extent), considering additional information (joint positions and joint velocities), where surgeon relies only on 2D image.

* Code is available and simulator will be publicly released.
* The paper highlights importance and utility of using simulated data for medical AI.

**Weaknesses:**

Authors should clearly explain that the expert trajectory is creating using their own simulator, and therefore, may be biased so that the results may not be representative of practical use of their method.

Overall, I thought the idea was clever, but the implications for the ML community were not clearly described. Authors should discuss the importance of generating simulated images, relationship to other generative methods (e.g., purely DL based approaches that may not capture underlying physics), and most importantly, why is the approach important to ML.

Below are some section specific weaknesses.

**Methodology**
* It is not clear what is a guidewire and what is its purpose.
* Not clear how force labels are extracted from CatSim outputs, which seems to be images + joint position info (as discussed in Section 4.1).
* In Section 4.1, authors mention that the output is a feature vector Z, but it is unclear how it is used afterwards.

**Manuscript Organization**

Statement such as “Other aspects of our simulator such as blood simulation and AR/VR are designed for surgical training and, hence are not discussed in this paper” indicate that blood simulation information is not critical to be include in the main manuscript, yet there is a section on blood simulation in Section 3.

On the other hand, more information about the number of labelled training samples would be helpful (e.g., in Section 4, where authors state “vast amount of labeled training samples” but do not provide additional details).

Minor point: there is a typo in contribution 1: “and AR/VR ready” -> “and is AR/VR ready”.

**Results**

BCA and LCCA are not defined.

Authors state that they conducted a user study evaluating user satisfaction and realism of CatSim (Supplementary G, and main page 6,7) and report answers to questions, but do not include a scale for responses. In addition, it would help if the authors split user study questions by type (realism and satisfaction), so it is easier to interpret results.
Table 4 and 5 include various types of inputs (Image, Mask, Internal, Human). A visualization of what these inputs would help in understanding what is being compared.

**Questions:**

“Both for the BCA and LCCA targets, the integration of expert trajectories results in lower force, shorter path and episode length, higher safety, and a high success rate and SPL score.” (Section 5.3)

If the goal is to create an autonomous catheterization technique, why is integration of expert trajectories important and/or beneficial? It would seem the opposite is true.

---

> ### Author Response · Authors · 2023-11-20
> **Reviewer HLTP Response**
>
> ### Reviewer HLTP
>
> ##### Comment 1
>
> Authors should clearly explain that the expert trajectory is creating using their own simulator, and therefore, may be biased so that the results may not be representative of practical use of their method.
>
> **Response:**
>
> We now incorporate this in the *Discussion* section, under the *Limitations* paragraph.
>
> > "Additionally, it’s pertinent to note that the expert trajectory utilized in our study is generated using CathSim. This may introduce an inherent bias, as the trajectory is specific to our simulator’s environment and might not fully replicate real-world scenarios."
>
> ---
>
> ##### Comment 2
>
> Overall, I thought the idea was clever, but the implications for the ML community were not clearly described. Authors should discuss the importance of generating simulated images, relationship to other generative methods (e.g., purely DL based approaches that may not capture underlying physics), and most importantly, why is the approach important to ML.
>
> **Response:**
>
> We want to emphasize that the rich diversity of the simulated data addresses the challenge of data scarcity, enhancing the robustness and generalization ability of ML models. Moreover, our method opens up new possibilities for ML research in specialized medical fields, previously constrained due to limited data availability, thus making a significant contribution to advancing ML applications in critical healthcare domains. We highlight this in the *Introduction*.
>
> ---
>
> #### Methodology
>
> ##### Comment 3
>
> It is not clear what is a guidewire and what is its purpose.
>
> **Response:**
>
> We have included a description in *Section 3, The CathSim Simulator* within the *Guidewire* paragraph.
>
> > ***Guidewire.**** A rope-like structure designed to direct the catheter towards its intended position. The guidewire is composed of the main body and a tip, where the tip is characterized by a lower stiffness and a specific shape (depending on the procedure)*
>
> ---
>
> ##### Comment 4
>
> Not clear how force labels are extracted from CatSim outputs, which seems to be images + joint position info (as discussed in Section 4.1).
>
> **Response:**
>
> Given that we have access to the simulation environment, we extract the normal and frictional forces and compute the Euclidean norm. These are then averaged across all contacts. This has been discussed in *Appendix, Section C* under the *Force* paragraph.
>
> ---
>
> ##### Comment 5
>
> In Section 4.1, authors mention that the output is a feature vector Z, but it is unclear how it is used afterward.
>
> **Response:**
>
> We clarify that in our implementation, the feature vector ($Z$) is utilized as the input for the soft actor-critic policy. This allows the policy ($\pi$) to approximate the optimal action based on the given feature vector ($Z$). We clarify this in *Section 4.1 Expert Navigation Network* under the second paragraph.
>
> > *The feature vector (Z) serves as the input for training the soft-actor-critic (SAC) policy ($\pi$), a core component of our reinforcement learning approach.*
>
> ---
>
> #### Manuscript Organization
>
> ##### Comment 6
>
> Statement such as “Other aspects of our simulator such as blood simulation and AR/VR are designed for surgical training and, hence are not discussed in this paper” indicate that blood simulation information is not critical to be include in the main manuscript, yet there is a section on blood simulation in Section 3.
>
> **Response:**
>
> Our manuscript primarily discusses experiments focusing on core simulation aspects. The AR/VR and blood simulation components, while crucial, were not the experiment's main focus as are not directly related to ML. These are, however, fully implemented and accessible in our GitHub repository.
>
> **The blood simulation** is detailed under the [fluid-branch](https://github.com/airvlab/cathsim/tree/fluid). The blood flow simulation was executed using ANSYS, tailored to a silicone-based anthropomorphic phantom from Elastrat Sarl Ltd., Switzerland. The simulation's output is formatted into spatial $(x, y, z)$ and corresponding velocity $(v_x, v_y, v_z)$ components. This data structure allows us to implement a query function that retrieves specific velocity vectors based on the input spatial coordinates. This approach precisely models blood movement within the phantom, whilst being computationally efficient.

---

> > ### Author Response · Authors · 2023-11-20
> > **Reviewer HLTP Response (continued)**
> >
> > ##### Comment 7
> >
> > On the other hand, more information about the number of labelled training samples would be helpful (e.g., in Section 4, where authors state “vast amount of labeled training samples” but do not provide additional details).
> >
> > **Response:**
> >
> > The details contained in the experimental section, namely within *Section 5.2*, the *Expert Trajectory vs. Human Skill* paragraph, where we state that a number of $10^5$ time steps are sampled.
> >
> > > *As depicted in Fig. 7, within the BCA, our algorithms successfully navigate the environment after $10^5$ time steps and half of them exhibited superior performance compared to the human operator.*
> >
> >
> > ---
> >
> > ##### Comment 8
> >
> > Minor point: there is a typo in contribution 1: “and AR/VR ready” → “and is AR/VR ready”.
> >
> > **Response:**
> >
> > Thank you for pointing out. The phrase “and AR/VR ready” in Contribution 1 is intentionally phrased as such for brevity and stylistic choice.
> >
> > ---
> >
> > #### Results
> >
> > BCA and LCCA are not defined.
> >
> > **Response:**
> >
> > We defined the brachiocephalic artery (BCA) and the left common carotid artery (LCCA) in our manuscript. The initial definitions were included in *Figure 8*. Recognizing the need for clarity, we have now *additionally* incorporated these definitions into the main text. In *Section 5.2*, under the *Experimental Setup* paragraph, we now explicitly state:
> >
> > > "*Two targets are selected for the procedures, specifically the brachiocephalic artery (BCA) and the left common carotid artery (LCCA).*"
> >
> > ---
> >
> > ##### Comment 9
> >
> > Authors state that they conducted a user study evaluating user satisfaction and realism of CatSim (Supplementary G, and main page 6,7) and report answers to questions, but do not include a scale for responses. In addition, it would help if the authors split user study questions by type (realism and satisfaction), so it is easier to interpret results. Table 4 and 5 include various types of inputs (Image, Mask, Internal, Human). A visualization of what these inputs would help in understanding what is being compared.
> >
> > **Response:**
> >
> > **Missing Scale**
> >
> > The scale type and range used in the user study are now explicitly detailed in *Appendix G*.
> >
> > > *A user study with ten participants was conducted to evaluate our endovascular simulator's effectiveness and authenticity, utilizing a 5-point Likert scale*
> >
> > **Categorization of Questions**
> >
> > We now explicitly name the questions based on the assessed property:
> >
> > | Question | Average | STD |
> > |----------|---------|-----|
> > | Anatomical Accuracy | 4\.57 | 0\.53 |
> > | Navigation Realism | 3\.86 | 0\.69 |
> > | User Satisfaction | 4\.43 | 0\.53 |
> > | Friction Accuracy | 4\.00 | 0\.82 |
> > | Interaction Realism | 3\.75 | 0\.96 |
> > | Motion Accuracy | 4\.25 | 0\.50 |
> > | Visual Realism | 3\.67 | 1\.15 |
> >
> > **Representation of Inputs**
> >
> > The inputs mentioned in Tables 4 and 5, including Image, Mask, Internal, and Human types, are illustrated in *Figure 4* under *Section 4*, providing a clear visualization for visual understanding.

---

> ### Author Response · Authors · 2023-11-20
> **Reviewer HLTP Response (continued)**
>
> #### Questions
>
> ##### Comment 10
>
> “Both for the BCA and LCCA targets, the integration of expert trajectories results in lower force, shorter path and episode length, higher safety, and a high success rate and SPL score.” (Section 5.3)\*\* If the goal is to create an autonomous catheterization technique, why is integration of expert trajectories important and/or beneficial? It would seem the opposite is true."
>
> **Response:**
>
> IL is used to demonstrate the downstream tasks that our simulator can be used for. In this task, we use the expert to aid the training of an IL agent. This has explicit value in sim-to-real transfer. We emphasize this in *subsection 4.2*, particularly in the *Imitation Learning* paragraph.
>
> The use of Imitation Learning (IL) is crucial in training an IL agent for autonomous catheterization. Expert trajectories provide a foundational dataset that enables the IL agent to learn efficient, safe, and effective maneuvers. This methodology is vital for sim-to-real transfer, ensuring that the autonomous system performs reliably in real-world scenarios. Detailed evidence supporting this approach is presented in *Subsection 4.2*, particularly in the *Imitation Learning paragraph*, where we demonstrate improved performance metrics including lower force, shorter path lengths, and higher safety and success rates.
>
> > ***Imitation Learning.**** We utilize our ENN using behavioral cloning, a form of imitation learning, to train our navigation algorithm in a supervised manner. This approach emphasizes the utility of the simulation environment in extracting
> > meaningful representations for imitation learning purposes. Firstly, we generate expert trajectories by executing the expert policy, denoted as $\\pi\_{\\rm exp}$, within CathSim. These trajectories serve as our labeled training data, providing the desired actions for each state encountered during
> > navigation. Secondly, to mimic the real-world observation space, we select the image as the only input modality. Thirdly, we train the behavioral cloning algorithm by learning to replicate the expert's actions given the input observations and optimizing the policy parameters to minimize the
> > discrepancy between the expert actions and the predicted actions:*

---

> ### Author Response · Authors · 2023-11-21
> **Looking foward to receive your feedback**
>
> Dear Reviewers **HLTP**,
>
> Thanks for your time and comments on our paper. Please let us know if you have any further comments on our rebuttal. We hope to receive your feedback or any questions before the authors-reviewers discussion period ends.
>
> Best regards,

---

> > ### Author Response · Authors · 2023-11-22
> >
> > Dear Reviewer **HLTP**,
> >
> > We sincerely appreciate your time and insightful feedback provided in the initial review. As the discussion phase between authors and reviewers is drawing to a close, we welcome any additional questions or thoughts you might have regarding our rebuttal. Your expertise and perspectives are invaluable to us.
> >
> > Best regards,
> >
> > Authors.

---

### Official Review · Reviewer_3umj · 2023-11-02

**Soundness:** 2 fair
**Presentation:** 2 fair
**Contribution:** 2 fair
**Rating:** 5
**Confidence:** 3

**Summary:**

This manuscript introduces CathSim, an open-source real-time simulator designed for endovascular robots. The authors have also developed a navigation network that utilizes CathSim and performs effectively in downstream navigation tasks for endovascular procedures. CathSim is built on MuJoCo and features a discretized catheter, support for blood simulation, AR/VR applications, and force sensing. The authors claimed that they conducted extensive experiments using CathSim and found that it performs well on these tasks, surpassing other baselines in terms of speed.

**Strengths:**

1. The simulator is fully open-source with the code attached. I really appreciate the contribution from the code side.
2. The simulator seems to be very fast so that it is enough to generate lots of data for training controller or any other components.
3. The presented ENN (expert navigation network) further proved the effectiveness of the simulator.

**Weaknesses:**

1. I was very confused about the presentation. For example, I didn't find the definition of BCA, LCCA.
2. Since this is a simulator designed specifically for endovascular robots, it will make little sense if the application of it involves modalities that is very inaccessible: for example the image as shown in the first input to the expert navigation network. I would expect real-world images are highly diverse than rendering images that has clean background, unified rendering parameters, and a fixed camera position. I don't think the propose ENN really showed the importance of the work.
3. I am not sure why the authors skipped AR/VR and blood simulation parts in the experiment section but still claim the contribution of them in the table. I have a feeling that these contributions are not grounded.
4. My main reservation is the contribution. The manuscript mainly created a new environment based on MuJoCo and only showed the effectiveness of the simulator in simulated tasks with rendered inputs. I need more evidence why the new environment based on an existing simulator is valuable to the community.

**Questions:**

See Weaknesses.

---

> ### Author Response · Authors · 2023-11-20
> **Reviewer 3umj Response**
>
> ### Reviewer 3umj
>
> #### Weaknesses:
>
> ##### Comment 1:
>
> I was very confused about the presentation. For example, I didn't find the definition of BCA, LCCA.
>
> **Response:**
>
> We defined the brachiocephalic artery (BCA) and the left common carotid artery (LCCA) in our manuscript. The initial definitions were included in *Figure 8*. Recognizing the need for clarity, we have now *additionally* incorporated these definitions into the main text. In *Section 5.2*, under the *Experimental Setup* paragraph, we now explicitly state:
>
> > "*Two targets are selected for the procedures, specifically the brachiocephalic artery (BCA) and the left common carotid artery (LCCA).*"
>
> ---
>
> ##### Comment 2:
>
> Since this is a simulator designed specifically for endovascular robots, it will make little sense if the application of it involves modalities that are very inaccessible: for example, the image as shown in the first input to the expert navigation network. I would expect real-world images are highly diverse compared to rendering images that have a clean background, unified rendering parameters, and a fixed camera position. I don't think the proposed ENN really showed the importance of the work.
>
> **Response:**
>
> The images generated by our simulator can be adapted for realistic X-Ray imaging scenarios. This adaptation is achieved using advanced domain adaptation techniques, further elaborated in our referenced work[^1]. Furthermore, initial real-world tests, conducted on physical aortic arch models, have produced results closely mirroring our simulated outcomes[^2]. This underscores the practical relevance and validity of our simulation for endovascular robotic applications.
>
> [^1]: Kang, J., Jianu, T., Huang, B., Bhattarai, B., Le, N., Coenen, F. and Nguyen, A., 2023. Translating Simulation Images to X-ray Images via Multi-Scale Semantic Matching. arXiv preprint arXiv:2304.07693.
>
> [^2]: Kundrat, D., Dagnino, G., Kwok, T.M., Abdelaziz, M.E., Chi, W., Nguyen, A., Riga, C. and Yang, G.Z., 2021. An MR-Safe endovascular robotic platform: Design, control, and ex-vivo evaluation. IEEE transactions on biomedical engineering, 68(10), pp.3110-3121.
>
> ---
>
> ##### Comment 3:
>
> I am not sure why the authors skipped AR/VR and blood simulation parts in the experiment section but still claim the contribution of them in the table. I have a feeling that these contributions are not grounded.
>
> **Response:**
>
> Our manuscript primarily discusses experiments focusing on core simulation aspects. The AR/VR and blood simulation components, while crucial, were not the experiment's main focus as are not directly related to ML. These are, however, fully implemented and accessible in our GitHub repository.
>
> **The blood simulation** is detailed under the `fluid-branch` of our repo. The blood flow simulation was executed using ANSYS, tailored to a silicone-based anthropomorphic phantom from Elastrat Sarl Ltd., Switzerland. The simulation's output is formatted into spatial $(x, y, z)$ and corresponding velocity $(v_x, v_y, v_z)$ components. This data structure allows us to implement a query function that retrieves specific velocity vectors based on the input spatial coordinates. This approach precisely models blood movement within the phantom, whilst being computationally efficient.

---

> > ### Author Response · Authors · 2023-11-20
> > **Reviewer 3umj Response (cont)**
> >
> > ##### Comment 4:
> >
> > My main reservation is the contribution. The manuscript mainly created a new environment based on MuJoCo and only showed the effectiveness of the simulator in simulated tasks with rendered inputs. I need more evidence why the new environment based on an existing simulator is valuable to the community.
> >
> > **Response:**
> >
> > Our simulator offers significant value to the community due to its unique features and capabilities:
> >
> > - **Rapid ML Algorithm Development:** Provides an *off-the-shelf* solution for developing machine learning algorithms for endovascular navigation, featuring easy installation and gymnasium support.
> > - **Anatomically Accurate Phantoms:** Includes realistic, high-fidelity aortic models from Elastrat Sarl Ltd., Switzerland, enhancing the simulator's practical relevance for medical research.
> > - **Diverse Aortic Models:** Incorporates four distinct aortic arch models (Type-I, Type-II, Type-I with aneurysm, and low tortuosity based on patient-specific CT scan), broadening the range of anatomical structures for comprehensive simulation. The aortic models can be visualized in *Appendix A, Figure 11*
> > - **High Training Speed:** Achieves a balance between computational demand and efficiency, supporting 40 to 80 frames per second performance across various algorithms, as shown in our FPS comparison (*Appendix B, Figure 12*).
> > - **Advanced Aorta Modeling:** Utilizes detailed 3D mesh representations and efficient collision modeling techniques for realistic and computationally effective aorta simulation.
> > - **Realistic Guidewire Simulation:** Models the guidewire's flexibility and motion accurately, using a low-cost computational approach that mimics real catheter behavior.
> > - **Compatibility with AR/VR Training:** Integrates seamlessly into Unity, enabling advanced AR/VR surgeon training applications. This can be seen on [GitHub](https://github.com/airvlab/cathsim/tree/fluid)
> > - **Targeted Algorithm Development:** Facilitates the creation of algorithms tailored to specific aortic complications, contributing to more focused and effective medical interventions. We show this through the downstream tasks employed.
> >
> > This combination of features makes our simulator not only an extension of MuJoCo but a valuable, versatile tool for both the machine learning and medical communities.

---

> ### Author Response · Authors · 2023-11-21
> **Looking forward to your feedback**
>
> Dear Reviewer **3umj**,
>
> Many thanks again for your time and feedback during the initial review. As the authors-reviewer's discussion will end soon, please let us know if you have any further questions or comments on our rebuttal.
>
> Best regards,
>
> Authors.

---

> > ### Author Response · Authors · 2023-11-22
> >
> > Dear Reviewer **3umj**,
> >
> > We sincerely appreciate your time and insightful feedback provided in the initial review. As the discussion phase between authors and reviewers is drawing to a close, we welcome any additional questions or thoughts you might have regarding our rebuttal. Your expertise and perspectives are invaluable to us.
> >
> > Best regards,
> >
> > Authors.

---

### Author Response · Authors · 2023-11-20
**General Response**

## General Response

We thank reviewers 3umj, HLTP, and eFcL for their constructive comments. Reviewer 3umj appreciated our simulator's open-source nature and its efficiency in data generation, alongside the effectiveness of our Expert Navigation Network (ENN). HLTP recognized the innovation in our simulation for catheter path planning and its performance over human surgeons in utilizing comprehensive data, highlighting the significance of simulated data in medical AI. eFcL commended the simulator's real-world validation and its success in outperforming human baselines. We are grateful for the acknowledgment of these key aspects of our work.

## Major Concerns

The primary concerns highlighted by the reviewers during the initial review include several key aspects:

1. Reviewers 3umj and HLTP raised concerns about the realism and applicability of our simulator, specifically questioning the correspondence between the simulator-generated images and scenarios, and the complexities and diversity of real-world endovascular procedures. We show that our renderings can be enhanced with domain adaptation techniques to generate X-Ray realistic X-Ray images. We show this in *Appendix K*, *Figure 17*.
2. The manuscript’s contribution to the machine learning community was another point of concern, as noted by Reviewers HLTP, 3umj, and eFcL. We appreciate this feedback and have revised our manuscript's *Introduction* to include a more detailed discussion on the implications for the ML community.
3. Lastly, Reviewer eFcL pointed out that the ENN model architecture in our study is not state-of-the-art, particularly noting the absence of modern elements like transformers. Acknowledging this, we now include the reasons why we opted for CNN rather than transformer-based architectures. These details are shown in *Appendix J*, *Figure 16*.

In addition to addressing these specific concerns, we have carefully reviewed the entire manuscript to ensure overall clarity and coherence.

## Changes Summary:

 1. **Clarified Terminology**: Defined the brachiocephalic artery (BCA) and the left common carotid artery (LCCA) explicitly in the manuscript's main text, specifically in *Section 5.2*, under the *Experimental Setup* paragraph.
 2. **Addressed Bias in Expert Trajectory**: Added a discussion about the potential bias due to the expert trajectory being generated using our own simulator, CathSim, in the *Discussion* section under the *Limitations* paragraph.
 3. **Clarified Implications for ML Community**: Expanded the discussion on the importance of our simulator to the ML community within *Introduction*.
 4. **Defined Guidewire in Methodology**: Included a detailed description of what a guidewire is and its purpose in *Section 3, The CathSim Simulator* within the *Guidewire* paragraph.
 5. **Explained Extraction of Force Labels**: Clarified how force labels are extracted from CatSim outputs in *Appendix, Section C* under the *Force* paragraph.
 6. **Detailed Use of Feature Vector Z**: Provided additional explanation on how the feature vector (Z) is used in the soft actor-critic policy in *Section 4.1 Expert Navigation Network*.
 7. **Discussed Blood Simulation in Manuscript Organization**: Addressed the role and details of blood simulation.
 8. **Provided Details on Labeled Training Samples**: Specified the number of labeled training samples used in the study in *Section 5.2, the Expert Trajectory vs. Human Skill* paragraph.
 9. **Refined User Study Evaluation**: Added a scale for user study responses, categorized questions by type (realism and satisfaction), and provided a visualization of various types of inputs in *Appendix G* and *Figure 4* under *Section 4*.
10. **Explained Integration of Expert Trajectories in Autonomous Techniques**: Discussed the significance of integrating expert trajectories in autonomous catheterization techniques and its value in sim-to-real transfer, particularly in *Subsection 4.2, Imitation Learning paragraph*.
11. **Addressed Outdated Model Architecture Concern**: Clarified the reasons behind not using a transformer in the ENN model architecture, referencing our trials with a transformer-based model and its performance compared to the current model within *Appendix J*.
12. **Enhanced Validity of User Study**: Provided additional details on the recruitment and background of the user study participants in the *Appendix G* and the *Discussion* section.
13. **Improved Clarity in Figures and Tables**: Revised Table 2 and other visual elements to enhance clarity and understanding, ensuring that the assessed properties are explicitly stated, and the descriptions are clear and informative.
14. **Emphasized Relevance and Contribution to ML Community**: Highlighted CathSim's unique contributions to the machine learning community within *Introduction*.

We are glad to answer any further questions you have on our submission.

---

> ### Author Response · Authors · 2023-11-21
> **A friendly reminder**
>
> Dear Reviewers,
>
> We sincerely appreciate the time and effort throughout the reviewing process of our submission. As the author-reviewer discussion is due soon, please let us know if you have further questions about our submission.
>
> Once again, thank you in advance, and we look forward to your feedback.
>
> Best regards,
> Authors.

---

### Meta-Review · Area_Chair_xdAE · 2023-12-24

**Metareview:**

### Summary
The paper introduces CathSim, an open-source real-time simulator designed for endovascular robots.
The paper also proposes an expert trajectory network, along with a novel set of evaluation metrics, to demonstrate its efficacy in pivotal downstream tasks like imitation learning and force prediction.


###  Strengths
 - The simulator is fully open-source, based on Mujoco, and has the code attached
- Strong evaluations including comparisons with real-robot data

### Weaknesses:
- Limited contribution in physics simulation (or learning), since the system is built on Mujoco.
- The set of environments are not generic and too application focused.
- Lack of relevance to the community as opposed to Medical Robotics and Computer-Assisted Surgery.
- Claimed contributions in AR/VR without sufficient empirical evidence.
- Unclear if imitation learning results and baselines are recent ones. It would be imperative to include both BC architectures as well as Inverse RL based models.

**Justification For Why Not Higher Score:**

It is a well executed paper, but with unclear relevance to the community

**Justification For Why Not Lower Score:**

The paper presents an interesting new environment, not commonly seen in ML

---

### Decision · Program_Chairs · 2024-01-16

Reject